# Recombinant Viral Vectors for Therapeutic Programming of Tumour Microenvironment: Advantages and Limitations

**DOI:** 10.3390/biomedicines10092142

**Published:** 2022-08-31

**Authors:** Karina Spunde, Ksenija Korotkaja, Anna Zajakina

**Affiliations:** Cancer Gene Therapy Group, Latvian Biomedical Research and Study Centre, Ratsupites Str. 1, k.1, LV-1067 Riga, Latvia

**Keywords:** viral vectors, tumour microenvironment, cancer immunotherapy, gene therapy

## Abstract

Viral vectors have been widely investigated as tools for cancer immunotherapy. Although many preclinical studies demonstrate significant virus-mediated tumour inhibition in synergy with immune checkpoint molecules and other drugs, the clinical success of viral vector applications in cancer therapy currently is limited. A number of challenges have to be solved to translate promising vectors to clinics. One of the key elements of successful virus-based cancer immunotherapy is the understanding of the tumour immune state and the development of vectors to modify the immunosuppressive tumour microenvironment (TME). Tumour-associated immune cells, as the main component of TME, support tumour progression through multiple pathways inducing resistance to treatment and promoting cancer cell escape mechanisms. In this review, we consider DNA and RNA virus vectors delivering immunomodulatory genes (cytokines, chemokines, co-stimulatory molecules, antibodies, etc.) and discuss how these viruses break an immunosuppressive cell development and switch TME to an immune-responsive “hot” state. We highlight the advantages and limitations of virus vectors for targeted therapeutic programming of tumour immune cell populations and tumour stroma, and propose future steps to establish viral vectors as a standard, efficient, safe, and non-toxic cancer immunotherapy approach that can complement other promising treatment strategies, e.g., checkpoint inhibitors, CAR-T, and advanced chemotherapeutics.

## 1. Introduction

Viral vectors have emerged as a promising approach to the treatment of cancer by selective lytic virus replication in tumours. The mechanism underlying the anti-tumour activity of viruses can mainly be categorized into two types. One is the selective destruction of tumour cells by oncolytic virus replication. This effect is influenced by the expression of virus cell surface receptors and the host cell’s antiviral response. The other mechanism of virus-mediated anti-tumour activity is associated with the induction of systemic anti-tumour immunity. It is becoming increasingly clear that antiviral immune responses can be efficiently exploited to induce innate and adaptive immunity against cancer cells.

Tumours create an environment that suppresses both innate and acquired immunity, and modulates the tumour microenvironment to prime the local immune response [1]. Despite genetic instability of cancer cells and the existence of multiple mechanisms to evade the immune response, changes in the tumour microenvironment (TME) may be common in many types of tumours, suggesting the possibility of therapeutic interventions into tumour-supporting mechanisms to treat different types of cancers. Tumour antigens released as a result of viral infection greatly enhance tumour epitope availability for activation of T cells [2,3]. The clinical development of innovative vector-based cancer therapeutics has been facilitated by the identification of tumour-specific neoantigens for which there is no self-tolerance and the approval of efficient immune checkpoint inhibitors (CPIs). Restoration of the patient’s immune response against their own malignancy by the use of CPIs is delivering promising results [4]. However, only a subset of patients currently benefits from CPIs therapy [5]. One of the causes is the T cells’ failure to infiltrate and recognize the tumour [6]. Expression of immuno-modulatory genes by the virus vector locally would help to stimulate antitumor immunity by the same time avoiding systemic toxicity issues.

While the use of vaccines for preventing infectious disease represents a story of great success, the development of effective cancer vaccines revealed to be more difficult, primarily due to the immune-suppressive microenvironment established by cancer cells, low immunogenicity of autologous tumour-associated antigens (TAAs) and plasticity of cancer cells. Therapeutic cancer vaccines sporadically achieved clinical benefits, with only limited products being approved by the FDA as Sipuleucel-T in 2010 to treat asymptomatic or minimally symptomatic metastatic hormone-refractory prostate cancer [7]. Furthermore, many approaches have failed, including the melanoma-associated antigen 3 (MAGE-A3) Phase III MA-GRIT study, the largest ever cancer vaccine trial in non-small cell lung cancer (NSCLC) patients [8]. Nevertheless, as promising approach cancer vaccines such as TAAs expressing viruses or TAAs coated viral particles continue to enter clinical trials such as MAGE-A3 expressing viruses (NCT02285816), papillomavirus E6E7 antigen expressing viruses (NCT03618953) as well as innovative peptide-coated conditionally replicating adeno-virus - PeptiCRAd-1 (NCT05492682) [9]. Significant work has been done to improve vector-based cancer vaccines [10], however, TAA-based vaccines are out of the scope of this review.

Many of the oncolytic viruses currently being tested in clinical trials are attenuated versions of common human pathogens. They have been genetically engineered to further reduce their pathogenicity and to increase oncolytic potency and specificity for cancer tissue. Clinical trials include both RNA and DNA viral vector-based vaccines and cancer treatments. A virus with a double-stranded DNA genome is the most suitable candidate for such manipulations with greater genome stability and a lesser chance of hazardous mutations. DNA vectors include adenovirus, vaccinia virus, adeno-associated virus (AAV) and herpes simplex virus (HSV). AAV is used as a preferred vehicle for liver-specific gene delivery [11], adenoviruses and herpes simplex virus have been extensively engineered and have been first approved for use in cancer therapy [12]. Despite the many attractive properties of RNA viruses, their use is still restricted due to challenging genetic manipulation even with the availability of the present-day cutting-edge genetic technologies. Clinically explored RNA vectors include measles virus, vesicular stomatitis virus (VSV), poliovirus, reovirus together with lentivirus, γ-retrovirus primarily employed for cell transduction [13,14,15]. The complexity of the antiviral immunity in the case of cancer virotherapy is far from being understood, and its contribution to the efficacy of virus vector-based treatment will probably depend on the particular virus species, TME state, and vector-delivered therapeutic transgene. Here we provide an overview of the most promising, in our vision, types of recombinant DNA and RNA virus vectors developed for cancer gene therapy, and their use for the anti-cancer immune response stimulation and reprogramming of the tumour immune microenvironment. The vectors providing stable transgene integration (retroviruses and AAV vectors) are not covered by this review.

## 2. Important Considerations for Immunotherapy Vector Selection

The choice of the virus vector depends primarily on the therapy target. These could be cancer cells or host cells. For the infection of immune cells with the reprogramming purpose, latent virus infection without cell death is needed. The vector should be able to target the tissue/tumour of interest. Ideally, every particular tumour has to be evaluated for the therapeutic virus infection/replication efficiency, as practice shows that the tumour cell susceptibility to the virus vector could be rather individual and difficult to predict. It seems that there is a need to work out a reliable test protocol for personalised tumour infectivity evaluation for main virus vectors applied in clinics, as this is the crucial point for virus therapy efficacy. A second important issue that has to be assessed is the possible pre-existing immunity against the vector, which could influence the outcome of therapy significantly [16].

While most large vectors are designed to be administered locally, there should be potential to deliver them systemically. In general, a compromise between the efficacy of therapy and safety has to be considered. Intravenous injection provides a potential opportunity for the virus to infect all cancer cells, including distal metastases [17]. However, viral particles injected systemically may be neutralised by the host immune response before reaching the target cells. The intratumoral injection can ensure that virus particles reach the tumour directly, but due to the dense tumour extracellular matrix, the spread of virus particles outside the injection area is restricted [18]. Therefore, the use of replication-deficient virus vectors is safe, but less efficient regardless of the systemic or local administration applied, and the use of attenuated replication-competent virus type would be more advantageous in respect of the efficacy of therapy.

Preferentially, for cancer gene therapy the vector should not be able to integrate into the cells. It has to be evaluated also if the short-term expression of the heterologous protein is enough or if the long-term production of the protein is essential for an efficacious result. The ability of the vectors to be used efficiently in repeated administration regimens, and/or heterologous virus prime-boost treatment strategy would be an advantage.

## 3. Viral Vector Platforms for Cancer Therapy

The general features of vector platforms are summarised in Table 1.

### 3.1. DNA Virus Vectors

#### 3.1.1. Adenoviruses

More than 50% of the gene therapy clinical trials are related to recombinant adenoviruses (AdVs, or Ad), which represent a highly potent gene delivery platform today. AdVs are non-enveloped viruses with an icosahedral nucleocapsid caring a double-stranded DNA genome of up to 45 kb in length [19]. Human Adenoviruses target the respiratory tract, gastrointestinal tract, and brain. Most adenoviruses use the Coxsackie Adenovirus receptor (CAR) to infect cells, but some species (HAdV-B) enter the cell through binding to the complement regulatory protein CD46, which is ubiquitously expressed on many cancer cells [20]. Studies in animal models and clinical trials have demonstrated efficient infection and replication in lung mesothelioma, lung carcinoma, ovarian epithelial-like tumours, adenosquamous carcinoma, head and neck squamous cell carcinoma, colon carcinoma, colorectal adenocarcinoma, gastric adenocarcinoma, prostate carcinoma, pancreatic adenocarcinoma, melanoma, fibrosarcoma cells, glioblastoma, hepatocarcinoma cells, leukemic monocytes/macrophages [21,22]. Last generation helper-dependent adenovirus vectors (HDAds), also called “gutless” or fully deleted vectors, containing only inverted terminal repeats and the genome pack-aging signal, allow up to 36 kb insertions. Besides high capacity, these third-generation vectors possess also reduced immunogenicity [23]. AdV vectors have the following advantages: (i) high insert capacity (up to 36 kb) and transduction efficiency, both in dividing and non-dividing cells, (ii) episomal persistence in the host cell, (iii) broad tropism for different tissue targets, and (iv) well-established vector production system. The major bottlenecks in AdV vector application are related to pre-existing anti-viral immunity within the human population [24] and the high immunogenicity of its capsid proteins inducing robust adaptive immune responses against de novo synthesized viral and transgene products [19].

New generation vectors with improved biosafety profiles and reduced immunogenicity in humans were developed on the basis of human serotypes with low seroprevalence, such as HAd2, HAd26, and HAd35, as well as on non-human adenoviruses (e.g., ChAd3, ChAdOx1, CAdVs). Nevertheless, the ability of these vectors to induce an immune response has been shown to be less potent compared with the most commonly used HAd5.118 vector [14]. Replication-deficient derivatives of the vectors were generated by deletion of the E1 genome region, increasing the biosafety of the vectors, and E3 genome region, increasing the capacity of the insert size. The selective replication in tumour cells is achieved by the creation of conditionally replicative adenoviral vectors (CRAdV) lacking E1B-55K and 24-amino acid of the CR2 domain of E1A, because in that case, the virus can preferentially replicate in cancer cells, known to carry p53 mutation and an excessive amount of E2F transcription factor, which complement AdV replication. Moreover, the insertion of the tumour-specific promoters into the AdV vector can be used to control the initiation of transgene production [25].

ONYX-105 (dl1520) was the first CRAdV vector to enter clinical trials, which together with a similar replication-selective vector H101, reached commercial approval in China in 2008 [26]. Although the safety of ONYX-105 and H101 was confirmed, the therapeutic efficacy was relatively low, probably, due to inefficient vector replication in patient tumour tissues. Currently, the chimeric and tropism-modified AdV vectors armed with anti-cancer and immunomodulating genes are under extensive investigation [15].

#### 3.1.2. Poxviruses

Poxviruses comprise a large family of enveloped double-stranded DNA viruses replicating in the cytoplasm. Their genome varies from 130 to about 300 kb. The attenuated strains such as Modified Vaccinia Ankara (MVA) vectors accept more than 7.5 kb of foreign sequences [27], whereas other poxviruses can incorporate over 24,000 bp of foreign genes in their genome [28]. Viruses enter the cell via plasma membrane fusion and after entry into the cytosol DNA replication occurs and the progeny virions are assembled and released from the cells.

In terms of cancer therapy Vaccinia virus (VV) was used successfully in triple-negative breast cancer models, renal cell cancer, colorectal cancer, hepatocellular carcinoma, melanoma, and osteosarcoma. Vaccinia strains have been shown to possess natural tumour tropism [29,30]. Although the specific receptor has not been identified, the tumour targeting can be attributed to the fact that many of the hallmarks of cancer cells (e.g., blocks in apoptotic pathways, dysregulation of cell cycle control, and inhibition of innate immunity) represent favourable conditions for VV replication [31]. Furthermore, VV spread is dependent on epidermal growth factor receptor (EGFR) signalling, a pathway that is activated in most cancers [31]. Although the majority of solid tumour cell lines are susceptible to vaccinia infection, indeed, leukaemia and lymphoma cells were resistant to infection [32].

Advantages of using poxvirus as a vector include the large capacity for foreign DNA insertion; the relatively high level of transgene expression and ability to induce both humoral and cellular responses; wide host range and preferential targeting of cancer cells of some virus species; lack of genomic integration; nucleus independent cytoplasmic replication; the low prevalence of anti-vector immunity in the human population thanks to the interruption of vaccination against smallpox [14]; established protocols for clinical grade large-scale production of vectors. Replication deficient Poxvirus vectors encoding heterologous antigens have a lower ability to prime immune responses in humans than other viral vectors [33].

Another relevant advantage of the VV over other vectors is the high stability and resistance to complement and antibody neutralization upon systemic vector injection because the virus produces an additional protecting membrane coat [34,35,36]. This property enhances tumour delivery and the spread of oncolytic VV for cancer treatment [30,32,37].

#### 3.1.3. Herpesviruses

There are known nine herpesviruses, infecting humans [38]. Most promising vectors were developed on the basis of herpes simplex virus type 1 (HSV-1). Herpesviruses are enveloped viruses with double-stranded DNA genomes ranging from 120 to 250 kbp in size. The viruses replicate in the nucleus and can establish latent infection in specific tissues, allowing them to evade the host’s anti-virus immune response.

Depending on the virus type, herpesviruses possess broad cell tropism. Alphaherpesviruses (HSV1,2, Varicella zoster) primary infect mucoepithelial cells and use neurons for the latent phase. Beta- and gamma herpesviruses target epithelial cells, monocytes and lymphocytes. HSV-1 entry is mediated by Herpesvirus Entry Mediator—HVEM and nectin 1. HVEM is expressed in a wide variety of immune cells including T and B cells, dendritic cells, natural killer cells, macrophages, polymorphonuclear cells, and in other cell types such as neurons, epithelial cells and fibroblasts [39]. HSV-1 vectors were successfully applied in neuroblastoma, glioma, lung carcinoma, ovarian adenocarcinoma, colon carcinoma, and melanoma.

The ability to cross the blood-brain barrier to treat glioma, high stability of the genome, potent cytolytic capability, and well-established genome engineering platform make herpesvirus vectors an attractive class of anti-cancer therapeutics [40,41]. In recent years, a number of armed vectors have been developed to improve anti-tumour activity through a variety of mechanisms, or through combination with existing treatment strategies. T-VEC caring GM-CSF is the only viral vector approved for clinical use in the EU and US [42,43]. 

Deletion of neurovirulence factor ICP34.5 attenuates HSV viral pathogenicity allowing the virus to replicate selectively in tumours. Furthermore, the HSV1 vector carrying combined deletions in ICP34.5 and ICP6 (large subunit of ribonucleotide reductase)—G207, was engineered, allowing virus replication in human glioma cells. However, deletion of ICP34.5 attenuated virus replication in glioma cells. G207 backbone vector was further modified by deletion of the ICP47 gene. The resulting triple-mutated G47∆ vector showed improved efficiency in glioblastoma tumours in comparison to the parent G207 vector [41]. In addition, ICP47 deletion reduces virus-mediated suppression of antigen presentation. The G47∆-based therapy for glioblastoma-teserpaturev Delytac-recently received approval for treatment of GBM in Japan [44].

### 3.2. RNA Virus Vectors

#### 3.2.1. Rhabdovirus

Rhabdoviruses (RVs) are bullet-shaped enveloped viruses with a single-stranded negative-sense RNA genome of about 11–16 kb in length. The most important vector systems were established on the basis of Rabies virus (RBV) and Vesicular Stomatitis virus (VSV). The replication of these viruses relies on the viral RNA-dependent RNA polymerase, which is encapsidated together with the viral genome. The maximum packaging capacity of the vectors is about 6 kb. However, the most optimal foreign sequence insert should not exceed a 4 kb fragment, because larger inserts appear to reduce viral replication in animal models [45]. The major advantage of RVs vectors is related to their attenuated replicative capacity and the ability to be pseudotyped with heterologous virus surface proteins, modulating the virus tropism and anti-vector immunity.

Many studies have confirmed that RVs are able to selectively and efficiently replicate in tumour cells, and the virus exhibits strong antitumor effects in a variety of preclinical trials [46,47,48]. The systemic administration of VSV can also specifically target multiple tumours in the brain, invasive tumours, and systemic metastatic tumours, making VSV one of the most promising oncolytic virus vectors for the treatment of various types of tumours. Clinical samples from chronic and acute myelomonocytic leukaemias appear to be especially susceptible to VSV [49]. On the other hand, VSV infection of tumour-associated DCs reduced their viability and prevented their migration to the draining lymph nodes to prime a tumour-specific CD8 T cell response [50], representing a disadvantage of these vectors as a tool for tumour vaccine development.

Development of a new more attenuated or replication incompetent version of the RVs vectors is required to broaden the spectrum of their application.

#### 3.2.2. Alphaviruses

Alphaviruses are enveloped plus-strand RNA viruses belonging to the *Togaviridae* family with a genome size of about 12,000 nt [51]. The alphavirus expression vectors are based on non-pathogenic avirulent strains to guarantee a high level of biosafety. There are three types of alphaviruses which are commonly used as expression vectors for heterologous gene delivery [52]: Semliki Forest virus—SFV; Sindbis virus—SIN; and Venezuelan equine encephalitis virus—VEE. Recombinant virus particles based on replication-deficient vectors were successfully tested for transient immunomodulating gene expression in different types of tumours. Usually, the gene of interest replaces the subgenomic RNA segment encoding structural genes. Therefore, the vectors are replication-deficient because the structural genes are not expressed and the progeny virions are not produced (one round infection vectors). The use of translation enhancer sequence from the virus capsid protein provides a very high level of transgene expression in vitro and in vivo in murine tumour models [52,53]. The capacity of the replication-deficient alphaviral vectors is restricted by the size of the structural genes, which is about 5000 nt long.

The alphavirus-based vectors possess broad tissue tropism and can infect a wide variety of human cells including different types of tumour cells, lymphoid cells, neuronal and glial cells, muscle cells, etc. Upon systemic injection, the SIN vectors are capable of targeting tumours [54,55], whereas SFV possesses more broad distribution [56]. It was shown that replication-deficient SFV vectors cannot infect human bone-marrow-derived macrophages [57], at the same time, some vectors can target dendritic cells [58], representing a significant advantage for induction of systemic antitumour immune response.

The receptors of alphaviruses were recently identified: very low-density lipoprotein receptor (VLDL-R) and apolipoprotein E receptor 2 (ApoER2) [59]. The low-density lipoprotein receptor (LDLR) family is involved in the endocytosis of cholesterol-rich low-density lipoprotein and other cell signalling ligands, and has shown to be evolutionarily highly conserved between mosquitoes and humans, allowing alphavirus transmission by arthropods. ApoER2 is enriched in the brain [60] explaining the neuronal tropism of some alphaviruses, which were used to target brain tumours (glioblastoma models). VLDL-R is abundant in heart, skeletal muscle, ovary and kidney cells, its overexpression correlates with breast cancer progression and metastasis [61,62].

Alphaviral vectors are good candidates for cancer gene therapy due to their ability to mediate strong cytotoxic effects by inducing immunogenic cell death [63]. Other advantages include p53-independent replication, a low specific immune response against the vector itself, a lack of vector pre-immunity in the majority of the population, and a high virus titre production system [64]. The combination of alphaviral vectors with advanced chemotherapeutics and checkpoint inhibitors has recently become a promising strategy for cancer treatment.

#### 3.2.3. Arenaviruses

Mammalian arenaviruses are enveloped, bi-segmented (S- and L-segments) ambisense single-stranded RNA viruses [65]. Arenaviruses, including lymphocytic choriomeningitis virus (LCMV) and Pichinde virus (PICV), can directly infect APCs, dendritic cells and macrophages. It was shown that LCMV binds to α-dystroglycan on the surface of APCs [66]. Nevertheless, the replication cycle of arenaviruses is not lytic and does not lead to killing the infected cell [65], therefore the activation of APCs and triggering of a potent anti-tumour CTL response can be expected. The studies in animal tumour models confirmed that LCMV vectors preferentially infect DCs, monocytes, and macrophages, over B and T cells [67]. Arenavirus vectors can accommodate transgenes of up to 2000 bp [68]. Foreign genes are integrated into replicating arenavirus vectors by redistributing the viral genes from two to three genome segments with duplicated S-segments [69].

The neutralizing antibody response to LCMV is extraordinarily weak due to the “glycan shield” covering the outer globular domain of the viral envelope glycoprotein [70], representing an excellent advantage for vector re-administration. Furthermore, the rare pre-existing anti-vector immunity in the human population and the ability to elicit protective CD8^+^ T cell immunity represents an additional advantage for cancer therapy. Importantly, unlike replication-deficient vectors, attenuated replication competent LCMV (artLCMV) vector is able to target not only dendritic cells, but also lymphoid tissue stromal cells. The infection of stromal cells triggers the IL-33–ST2 alarmin pathway inducing a superior CTL response in tumours [68]. Novel intravenously administered, replication-competent, non-lytic arenavirus-based vectors that deliver tumour antigens to induce anti-cancer T cell responses are currently in clinical testing for the treatment of Papillomavirus-associated cancer [71].

#### 3.2.4. Enteroviruses

The genus Enterovirus represents a ubiquitous group of viruses, including polio-, rhino-, coxsackie-, echo-, and numerous enteroviruses [72,73], belonging to the Picornaviridae family. The non-enveloped particles possess a compact structure and small size (up to 28 nm). The virus genome is encoded by a positive-sense, single-stranded RNA molecule of about 7–9 kb in length limiting the genome modification capabilities [74]. Poliovirus-1 (PV-1) and coxsackieviruses (CV-B3, CV-B4, CV-A9) have been used to develop viral vectors [72]. Immunoglobulin-like receptor CD155 (also known as the poliovirus receptor), human cell adhesion molecules, e.g., coxsackie-adenovirus receptor (CAR), as well as Nectin-like molecule 5 and RGD motif of integrins, were identified as receptors for enteroviruses [72]. Many of these receptors are overexpressed on cancer cells, facilitating the tumour tropism of enteroviruses.

The vector system of poliovirus is based on the generation of a dicistronic cassette containing duplicated IRES: one IRES from PV and the other from a related rhinovirus for insert expression [75], or by insertion of artificial protease cleavage sites [72]. The replacement of capsid proteins results in the synthesis of replication-deficient vectors in the presence of a helper virus [76]. Enteroviruses possess tumour tropism, are not highly pathogenic to humans, and do not integrate into the human genome. Due to the global poliovirus vaccination, it is likely that the other enteroviruses besides PV will attract more attention as gene therapy vectors in the near future.

The natural neurotropism of poliovirus makes the PV vector a promising candidate for the treatment of CNS tumours, e.g., malignant glioma, which overexpresses the Necl-5, potentiating virus targeting [77]. Currently, one such vector, recombinant oncolytic poliovirus (PVS-RIPO), is under clinical evaluation for cancer therapy [77]. Furthermore, Coxsackievirus A9 (CV-A9) strain uses the cell surface integrins as receptors for virus entry by binding the RGD motif [78]. Therefore, the CV-A9-based vectors represent a promising tool to target tumours overexpressing αVβ3 integrin in anti-angiogenic therapy [79]. Although the enterovirus-based vectors possess tumour tropism and showed significant oncolytic properties, the small insert size (<1 kb for replication-competent vectors) and the insufficiently developed vector production system limits their clinical translation.

#### 3.2.5. Reoviruses

Reoviruses are non-enveloped 85 nm in diameter viruses containing a segmented double-stranded RNA genome of about 23.5 kb in ten segments [80]. Importantly, no significant RNA genomic changes were detected in persistently infected cultures, including in the case of attenuated reovirus strains [81].

Reovirus replication induces oncolytic effects in many solid and haematological tumours including lung, breast, ovarian, prostate, colorectal, pancreatic, glioma, melanoma, head and neck squamous cell carcinoma, myeloma, and both lymphoid and myeloid leukaemias [82]. The tumour targeting is mediated, probably, by the virus interaction with sialic acid junctional adhesion molecule A (JAM-A), which is overrepresented on the surface of many cancer cells. Preferential virus replication in cancer cells is promoted by the active Ras signalling pathway, as well as by the inhibition of antiviral innate immune response, PKR inactivation [83].

In clinical trials, the wild-type reovirus strains demonstrated tolerability and safety [82]. Currently, the work with armed reovirus vectors for the expression of therapeutic genes is under development. The plasmid-based reverse genetics strategies include the insertion of small genes in the S1, M1 and L1 RNA segments (500–1000 bp of a single gene) retaining the replication-competent phenotype [84,85].

#### 3.2.6. Paramyxoviruses

Paramyxoviruses are enveloped viruses carrying a negative-sense, single-stranded RNA genome of about 16 kb in length. Similar to other negative-sense RNA viruses (e.g., rhabdoviruses), reverse genetics systems are used for virus production [86]. Genetic modification usually is performed by introducing the independent transcription unit, or through the internal ribosomal entry site for foreign sequence expression.

*Measles virus (MV).* Virus particles of MV are pleomorphic ranging from 150 to 350 nm in size. MV vectors accept relatively large inserts (up to 6 kb) [87]. MV wild-type strains enter cells predominantly through the signal lymphocyte-activation molecule (SLAM or CD150), which is expressed on activated B- and T- lymphocytes, memory lymphocytes, dendritic cells, and immature thymocytes. However, attenuated MV particles are capable of using CD46 as a cell entry receptor, which is a regulator of complement activation abundantly presented on cancer cells [88]. Nectin-4, which is known also as Poliovirus receptor-related 4 (PVRL4), was identified as another MV receptor. This receptor is predominantly expressed in the respiratory epithelium [89].

Attenuated vaccine strains of MV have been selected and used successfully as a safe and efficient vaccine platform. In contrast to other RNA viruses, MV vaccine strains demonstrate high genetic stability [89]. Overexpression of CD46 was found in gastrointestinal, hepatocellular, colorectal, endometrial, cervical, ovarian, breast, renal, and lung carcinomas, also in leukaemias and multiple myeloma [88]. The virus envelope was modified to increase tumour specificity [90].

Attenuated MV is a promising oncolytic agent. It was shown that MV-infected cells induced pDC maturation with a strong production of IFN-α [91]. The potential bottleneck for MV vector therapy is the pre-existing immunity in the population. However, it was shown that low titres of anti-MV antibodies still allow achieving cellular immune response to immunization with MV vector [92]. MV vectors can be genetically modified (armed) to enhance therapeutic outcomes.

*Newcastle disease virus (NDV).* NDV, also known as avian orthoavulavirus 1, possesses a pleomorphic virus particle of 100 nm in diameter. NDV accepts at least 4.5 kb foreign genes with good stability [93]. The reverse genetic approach was applied for all three NDV pathotypes, and a single NDV vector was shown to be able to express up to three different foreign genes [94]. NDV is antigenically different from common human pathogens, therefore the vector pre-immunity is absent or very low. Other advantages of NDV vector include the ability to infect a wide variety of tumour types through binding to sialic acids on the tumour cell surface. Oncolytic NDV has completed a phase I/II clinical trial in patients with glioma, demonstrating safety and good tolerability results [95].

## 4. Immunogenic Tumour Cell Death

Therapeutic viruses induce tumour cell death through multiple pathways including apoptosis, necroptosis, pyroptosis and autophagy [63]. Some viruses induce immunogenic cell death (ICD) of cancerous cells together with the release of danger-associated molecular patterns (DAMPs), such as ATP, nuclear high mobility group box 1 (HMBG1), calreticulin (CRT) and heat shock proteins (HSP) 70 and 90 that promote an anti-tumour immune response (Figure 1) [96,97]. ATP acts as a “find-me” signal, CRT—“eat-me” signal for APCs, while HMGB1 and HSP70/90 activate immune cells. These signals promote the activation of dendritic cells (DCs) and, as a result, the release of pro-inflammatory cytokines such as IL-1β, IL-6, IL-12, and TNFα, and facilitate the engulfing of dying cancer cells following tumour-associated antigen (TAA) presentation to the T cells [98]. Thus, the induction of ICD is an important step for long-lasting tumour-specific immunity. ICD can be induced by intracellular pathogens, as well as by physical and chemical stress. It was found that Semliki Forest virus, Measles virus, Newcastle disease virus, Coxsackievirus B3 and CD40L-encoding oncolytic adenovirus induce ICD in cancer cells [63,99,100,101,102]. In contrast to viruses, chemotherapy and physical approach may result in an impaired immune response. Furthermore, cancer cells are more sensitive to viral infections due to defective IFN-I pathway that makes them susceptible to virotherapy [103]. Thus, ICD-inducing viruses are highly promising tools for the activation of immunosuppressive TME. On the other hand, antiviral immunity may reduce therapeutic efficacy.

## 5. Tumour Microenvironment

The tumour microenvironment (TME) consists of stroma cells including cancer-associated fibroblasts (CAFs), vascular endothelial cells and tumour-associated immune cells, as well as extracellular matrix (ECM) and soluble factors such as cytokines and chemokines. The main tumour-infiltrated immune cells are T cells, monocytes/macrophages, dendritic cells (DCs), myeloid-derived suppressor cells (MDSCs) and natural killer cells (NKs). The effect of virotherapy on TME is not clear as the TME may display distinct immunological status. Tumour immune microenvironment can be divided into (i) immunostimulatory or immunologically “hot” and (ii) immunosuppressing or “cold” (Figure 2). The “hot” TME consists of immunostimulating factors and activated immune cells [104]. Immunostimulating factors include nitric oxide (NO), IL-1β, IL-6, IL-12, TNFα, and IFNs. It is considered that “hot” TME supports therapeutic anti-tumour immune responses. On the other hand, “cold” TME comprises immunosuppressive molecules and is associated with cancer progression and a worse survival prognosis [105]. Immunosuppressive factors include IL-4, IL-10, IDO, COX2, EGF, HGF, and TGFβ. Viral vectors can be used to program TME into an immunologically “hot” state.

### 5.1. Tumour-Associated Macrophages

Tumour-associated macrophages (TAMs) stimulate tumour growth, induce angiogenesis and promote an immunosuppressive environment by secreting tumour-inducing cytokines and chemokines [106]. Patients with a high density of TAMs in TME have a worse prognosis for disease-free survival (DFS) than those with a low density [107,108]. In addition, the co-culture of cancer cells and macrophages increases the invasiveness of cancer cells [109,110]. The TAMs phenotype varies depending on the tumour type due to macrophage plasticity. Conventionally, two polarization states have been distinguished for macrophages: classically activated (M1) and alternatively activated macrophages (M2 or AAM). The pro-inflammatory M1 is involved in the Th1 immune response, has high antigen presentation capacity, express MHC II^hi^ and iNOS (inducible NO-synthase), and release inflammatory cytokines such as IFNγ, TNFα, IL-6, IL 12, and IL-23 [111]. M2 is an immunosuppressive CD206^+^Arginase-1^hi^ phenotype involved in the Th2 immune responses, clearance of extracellular pathogens, allergy reactions, and the repair and remodelling of injured tissues [106,112]. M2 TAMs produce a high amount of immunosuppressive and proangiogenic factors such as IL-10, arginase, transforming growth factor β (TGFβ), or vascular endothelial growth factors (VEGFs [113,114,115]. Thus, M1 is associated with immunostimulatory TME, while M2, with the immunosuppressive TME.

### 5.2. Dendritic Cells

Dendritic cells (DCs) are a heterogeneous population of myeloid cells with diverse appearance, ontogeny, and immunological characteristics. DCs are professional antigen-presenting cells (APCs) and, because of their unique ability to induce T cell responses, DCs are the most effective adaptive immune activators. Mature DCs (mDCs) express MHC II, costimulatory molecules CD80 and CD86 and secrete pro-inflammatory cytokines that prime T cells to regress the tumour and metastases. However, antigen presentation by immature DCs (iDCs) can lead to T cell tolerance as T cells become anergic, suppressive, or are simply deleted [116]. An increase in the number of tumour-infiltrating DCs has been related to improved survival [117].

### 5.3. Myeloid-Derived Suppressor Cells (MDSCs)

MDSCs promote angiogenesis, tumour invasion and metastases through different soluble factors [118,119]. They suppress immune responses by the activation of Tregs and the production of inhibitory factors such as IL-10, TGFβ, indoleamine-2,3-dioxygenase (IDO), and cyclooxygenase-2 (COX2) [120,121,122,123]. In vitro studies showed that MDSCs promote epithelial-mesenchymal transition (EMT) by the production of epidermal growth factor (EGF), hepatocyte growth factor (HGF) and TGFβ [124]. Polymorphonuclear (PMN-MDSC) and monocytic MDSCs (M-MDSC) are the two primary subpopulations of MDSCs. Because of the biochemical and functional heterogeneity of MDSC populations, several subtypes of MDSC have been isolated from different forms of cancer. The percentage of infiltrating M-MDSC and PMN-MDSC changes with tumour type and disease progression [125]. To our knowledge MDSCs are rarely targeted in cancer therapy, existing strategies include MDSCs depletion, blockade, activity inhibition and differentiation into mature myeloid cells [126,127]. MDSCs reprogramming using viral vectors could be more specific and effective than chemotherapy.

### 5.4. Natural Killer Cells

The anti-tumour immune response is triggered by natural killer cells (NK cells) that can induce cancer cell lysis. Although high levels of tumour infiltrating NK cells are linked to a better prognosis in several human solid tumours, the immunosuppressive TME inhibits their activity, favouring tumour growth [128]. NK cell subsets are distinguished by the expression of the CD56 and CD16 cell surface markers. CD56^dim^ CD16^+^ NK cells may directly kill other cells via activating death signals (TRAIL, FAS) or releasing perforins and granzymes. CD56^bright^ CD16^−^ NK cells, on the other hand, are less cytotoxic but more immunomodulatory releasing pro-inflammatory cytokines such as IL-1, IL-2, IL-12, IL-15, and IL-18 [129].

### 5.5. T Cells

T cells are diverse cell populations with different immunophenotypes which play an indispensable role in adaptive cancer immunity. The main T-cell subsets in the TME consist of Tregs, helper T cells (Th), and cytotoxic T cells (CTLs). These T-cell subsets are involved in different immune functions, such as cytotoxic immune response mediated by CTLs or immune suppression mediated by Tregs and type II Th (Th2) cells. Infiltration of CTLs and type I Th (Th1) in the tumour is associated with a “hot” TME and a favourable prognosis, as these cells secrete IFNγ and induce tumour cell lysis [130,131]. On the other hand, the infiltration of Tregs and Th2 cells is associated with immunosuppression and, therefore, a negative prognosis. Tregs can release TGF-β, IL-10, and IL-35 during tumour immune escape, which decreases antitumour immunity, and suppresses antigen presentation by DCs and CD4^+^ Th cell activity. Persistent antigenic stimulation in the tumour triggers the mechanism of T-cell exhaustion leading to the loss of CTL effector function. The exhausted tumour-specific T-cells do not express characteristic pro-inflammatory cytokines such as IL-2, TNFα, and IFNγ, but instead produce the inhibitory receptors such as programmed death 1 (PD-1) and cytotoxic T-lymphocyte associated protein 4 (CTLA4) [132]. In addition, another subgroup of non-functional effector cells in TME are the so-called anergic T cells—incompletely activated T cells that do not express co-stimulatory molecules, probably due to insufficient priming by DCs.

### 5.6. CAFs and Vascular Endothelial cells

Non-immune cells also play an important role in TME. The heterogenous cancer-associated fibroblast (CAF) population has been demonstrated to stimulate tumour growth, progression, and treatment resistance [133]. Studies have shown that CAFs are involved in immunomodulation and ECM remodelling leading to metastasis [134]. ECM stiffness positively correlates with tumour malignancy [135]. ECM remodelling was shown to reduce the infiltration of T-cells and macrophages [136]. Furthermore, CAFs produce tumour progression stimulating factors such as TGFβ, VEGF, IL-6, and HGF [137]. Both CAFs and vascular endothelial cells are involved in angiogenesis, ensuring the supply of nutrients, oxygen and other metabolic factors necessary for tumour growth [138,139].

## 6. Therapeutic Strategies for Therapeutic Programming of TME

The ability of the virus vector itself to induce antiviral responses through pattern recognition receptors (PRRs) and ICD, attracts lymphocytes and macrophages that become activated and likely to recognize tumour antigens released as a result of virus-mediated tumour cell lysis, thereby greatly enhancing tumour epitope availability for activation of T-helper cells. However, despite showing great potential, research and clinical experience have demonstrated that the use of viruses only as oncolytic agents is insufficient to achieve stable therapeutic responses. To potentiate immunotherapy, virus vectors delivering therapeutic genes for immunosuppressive TME reprogramming have to be exploited.

The local virus-based delivery of immune regulatory factors to break immunosuppressive TME and stimulate pro-inflammatory responses is promising due to the ability to ensure high intratumoral concentrations of active molecules without systemic toxicity. Recombinant viruses can help to reverse immunosuppression and restore a more favourable inflammatory TME, by producing different types of immune-stimulatory proteins and peptides, such as cytokines, chemokines, CPIs, soluble receptors, therapeutic antibodies, inhibitory peptides, enzymes and hormones as well as gene silencing short hairpin or interfering RNAs. Two types of therapeutic strategies can be considered: (i) programming of tumour-associated immune cells; and (ii) programming of tumour stroma (Figure 3).

Cytokines are immunomodulatory proteins, produced in nano-picomolar concentrations by immune cells and regulate the functional activities of target immune cells and tissues. The most essential cytokines in the anti-tumour response are IL-12, granulocyte-macrophage colony-stimulating factor GM-CSF, T cell growth factor IL-2, IL-2-related cytokines IL-15 and IL-21 and pro-inflammatory IFNγ and TNFα. Furthermore, Insufficient infiltration of effector lymphocytes into the tumour often correlates with the low efficacy of immunotherapies [140]. Recombinant viruses encoding chemokines, which attract effector cells, may improve anti-tumour immunotherapy. Important chemokines in the TME are CCL2, CCL5 (RANTES), CXCL9 and CXCL10, which attract Th1 cells and CTLs [141]. It was suggested that when DCs are activated by cognate CD4^+^ T cells, DCs start secreting chemokines CCL3 and CCL4 attracting CCR5-expressing CD8^+^ T cells, finally resulting in strong CTL responses [142]. The expression of co-stimulatory molecules important for priming specific immune cell populations, such as expressed by mature DCs CD80/86, also represents a promising and actively explored strategy [105].

To restore the immune control of the tumour and reverse immunosuppression, different immune cell populations can be (re)programmed by virus-encoded immunotherapy genes. Generally, these can be subdivided into APCs—including DCs and macrophages; and effector cells—NK cells and adaptive immunity representing T-cells.

### 6.1. Programming of Tumour-Associated Immune Cells (Strategy 1)

**APCs.** Stimulation of APCs migration to the tumour by expressing GM-CSF with recombinant virus vectors is widely explored and successfully applied using different virus vectors. Local GM-CSF production allows to avoid undesirable toxicity and keeps high intratumoural GM-CSF concentration to enhance the recruitment and activation of DCs to prime CTLs and start a systemic anti-tumour response. On the other hand, it was shown that the tumour cells produce GM-CSF to shape pro-tumour TME, promoting myelopoiesis and recruitment of the tumour supporting MDSCs [143]. However, in the case of ICD induced by virus vector, cancer cells release DAMPs that activate DCs, as well as pathogen-associated molecular patterns (PAMPs) triggering type I IFNs production which is essential for DCs-driven T cell responses to cancer. In particular, Toll-like receptors (TLR) activation such as TLR3,7,9 on macrophages and plasmacytoid DCs by viral nucleic acids results in secretion of type I IFNs, Th1 type cytokines (e.g., TNFα, IFNγ, IL-2) and expression of co-stimulatory molecules (e.g., CD80, CD86) as well as lymph node homing signal CCR7 [144]. Therefore, the simultaneous stimulation of Th1 response by the DAMPs and PAMPs released during viral infection is crucial for GM-CSF therapy through maturation and activation of APCs and prevention of MDSCs phenotype induction due to immunosuppressive TME.

Additionally, macrophage recruitment to the TME is regulated also by chemokine CCL2, which is induced by pro-inflammatory anti-viral signalling pathways. The infiltration of macrophages has been shown in several in vivo tumour models treated with oncolytic viruses [145]. PAMPs binding to TLRs can induce inhibition of M2-related signalling and leads to M1-associated gene transcription. Macrophages are suggested to detect viral nucleic acids by TLR3,7 and 9, while virus-infected tumour cells by TLR2/1. Therefore, the TLR agonists, such as unmethylated cytosine-phosphate-guanine (CpG) rich DNA and double-stranded RNA molecules produced during virus vector replication, provide additional stimulation of the Th1 response.

Another approach is the activation of DCs through the CD40–CD40L pathway resulting in increased expression of MHC, co-stimulatory, and adhesion molecules, induction of Th1-type immune responses and T cell activation and migration into the TME. CD40 is a member of the tumour necrosis factor (TNF)/tumour necrosis factor receptor (TNFR) family, expressed on macrophages and DCs, which includes also 4-1BB (CD137), and CD27 [146]. 4-1BB is expressed on B cells, macrophages, DCs, as well as on activated T and NK cells, whereas its ligand 4-1BBL is expressed on DCs and macrophages. Subsequently, CD40 ligand (CD40L) is expressed by CD4^+^ T helper cells and binding to CD40 on DCs promotes DCs maturation. Thus, the expression of CD40L from the virus vector can provide additional DCs stimulation and T-cell priming [146]. A very promising virus vector candidate to express co-stimulatory molecules (CD80/CD86) is LCMV, which itself activates APCs and triggers a robust CD8^+^ T cell response to viral antigens [147]. Importantly, the infection of arenaviruses is not lytic it does not destroy the infected cells and the immune response against the virus is negligible [147]. Cancer cells avoid phagocytosis by expressing surface molecules such as CD47, which binds to the macrophage signal regulatory protein α (SIRPα) inhibiting IgG-mediated phagocytosis and integrin activation [148].

**T-cells, NK cells.** IL-12 is the most potent Th1 type cytokine produced by APCs, known to induce CTL and NK cell anti-tumour response. The development of strategies for efficient local IL-12 delivery to the tumour to avoid its systemic toxicity is of increasing interest. The targets of IL-12 immunotherapy are immune cells within the tumour including activated but exhausted T cells, NK cells, TAMs, and MDSCs. IL-12 increases the activation and cytolytic capacity of CTLs and NK cells and induces the production of IFNγ. IL-12-stimulated T cells expressed lower levels of programmed death 1 (PD-1) and higher levels of IFNγ and IL-2 compared to IFNα-stimulated T cells [149]. IL-12 has been shown to modulate and alter the suppressive activities of tumour-associated MDSCs [150], Therefore, maximizing the amount of IL-12 that reaches the tumour seems critical for a robust antitumor response.

IL-15 and IL-18, have shown promise as immunotherapeutic cytokines able to enhance both NK and T-cell responses [151]. IL-21 also is a potent inducer of T cell activation in vivo and can inhibit the development of suppressive Treg cells. It is necessary for the maturation, activation, and cytolytic potentiation of NK and NKT (natural killer T) cells. Additionally, it suppresses angiogenesis by reducing the expression of VEGF receptor 1 (R1) in endothelial cells [152].

The binding of peptide-loaded MHCII on APCs to antigen-specific T-cell receptor (TCR) and following engagement of co-stimulatory molecules CD80 (B7.1)()/CD86(B7.2) are necessary steps for T helper cells and subsequent effector CTLs activation, which are the main cells responsible for the removal of cancer cells [153]. CD28 is a T-cell surface receptor that binds to CD80 (B7.1) and CD86 (B7.2) on APCs triggering cell-mediated immune responses. A T cell receptor CD28 binding to B7.1 co-stimulatory molecule expressed by APC provides a potent activation signal resulting in the production of IL-2 and related cytokines, enhanced expression of CD25 (IL-2Rα), and inhibition of activation-induced cell death in the T cell.

Both, CD28 and cytotoxic T-lymphocyte associated protein 4 (CTLA-4) are two receptors that recognize the same ligands (B7 molecules) but have opposite functional effects on T-cell activation [154]. Whereas CTLA-4 is induced following T cell activation, CD28 is constitutively expressed on effector T cells. CTLA-4 is expressed constitutively by Tregs and is upregulated on the surface of CTLs during the early stages of activation, downregulating T-cell responses. Programmed death ligand 1 (PD-L1) and CTLA-4 antibodies (CPIs) are being actively tested in clinical trials, and also corresponding engineered antibodies expressed by virus vectors are studied by many research groups. CPIs showed very promising results in cancer therapy. Antagonizing PD-L1 is well known to restore PD-1/PD-L1-mediated T cell exhaustion and improve significantly the anti-tumour immune response [155]. PD-L1 is mainly expressed by APCs to avoid excessive tissue damage by activated CTLs expressing in turn corresponding PD-1 receptors. This inhibitory mechanism is usually exploited by the tumour as PD-L1 is upregulated in many cancers helping to escape immune attack.

Additionally, promoting the recruitment of tumour-specific Th1 effector cells and tumour infiltration with circulating anti-tumour CTLs is significant for efficient tumour immunotherapy. The chemokine CCL5 (RANTES) is produced by both APCs and activated T lymphocytes and is a broad chemoattractant, it binds to CCR1, CCR3, and CCR5 receptors expressed by T cells, macrophages and NK cells [141]. CCL5 was associated mainly with poor prognosis however, a combination of pro-inflammatory chemokine with viral infection can be a promising approach for the induction of tumour-specific immunity [156].

### 6.2. Programming of Tumour Stroma and Vasculature (Strategy 2)

The important complementary therapeutic strategy is to disrupt the dissemination of cancer cells by inhibiting the development of blood and lymphatic circulatory systems in tumours by blocking the activity of CAFs and vascular endothelial cells, as well as tumour cells, secreted angiogenesis and ECM stimulatory factors (Figure 3, strategy 2). Therefore, delivery of antiangiogenic and ECM modulating enzymes or hormones is promising, but still a less exploited option to reprogramme the TME. Tumour cells secret various pro-fibrotic growth factors and inflammatory factors such as TGFα, TGFβ, fibroblast growth factor (FGF)-2, platelet-derived growth factor (PDGF) and EGF. The ECM in solid tumours differs from that in normal organs, showing the accumulation of significant amounts of collagens, fibronectin, elastin, and laminins [157,158]. In addition, some cancers, such as pancreatic ductal adenocarcinomas (PDACs), are highly producing hyaluronan able to accumulate a large amount of water resulting in increased interstitial fluid pressure in the tumour [159]. To modulate tumour ECM and decrease interstitial tumour pressure, which impairs infiltration of immune cells and therapeutic virus spread, the collagen level regulating peptide hormone relaxin and enzymes, e.g., hyaluronidase, can be delivered by the virus vector. However, on the other hand, the probability of potential enhanced tumour cell invasion due to excessive proteolytic degradation of ECM should be considered.

TGFβ plays a central role in inducing immune tolerance in the tumour microenvironment and is expressed by tumour cells, CAFs as well as immunosuppressive macrophages. Moreover, a hypoxic condition in the tumour induces upregulation of VEGF to recruit endothelial cells and build new tumour vessels. VEGF is a pro-angiogenic protein shown to inhibit the proliferation and maturation of T cells and DCs. Therefore, downregulation of VEGF expression could help to restore anti-tumour immunity [160]. The antibodies, soluble receptors and inhibiting peptides, as well as short hairpin RNA molecules, can be expressed by virus vectors with the aim to block immunosuppressive factors in TME such as VEGF, TGF-β and, potentially, IL-10.

### 6.3. Simultaneous Targeting of Different Immune Cell Sub-Sets and/or Stroma (Combination of Two Strategies)

To improve the efficacy of therapy combined targeting of different immune populations and tumour stroma by expression of multiple immune-stimulatory genes working synergistically may be employed. With an aim to stimulate antigen presentation and T-cell trafficking, multiple cytokines can be expressed in combination. First of all, a potent GM-CSF and IL-12 combination is an example of APCs and T-cell stimulation. The use of local anti-TGFβ therapy and ECM regulating enzymes and hormones would improve the efficiency of the immune-stimulating therapy response by promoting virus vector spread and immune cell infiltration.

It is important to combine together not only immune response and cell trafficking stimulating cytokines. Simultaneous removal of T cell exhaustion by inhibiting PD1/PD1-L and CTLA-4 signalling and T cell anergy by expression of co-stimulatory molecules, such as CD-80, is crucial for induction of specific systemic immune response. Another underexplored option is the local expression of anti-IL-10 inhibiting molecules, which can help to reduce immunosuppressive TME pressure to the infiltrating immune cells. However, it would be difficult to assess the specific anti-tumour mechanism and expected bottlenecks during such complex therapy in case of suboptimal results.

## 7. Viral Vectors Used for TME Programming

### 7.1. Promotion of Professional APCs

The infection by the oncolytic virus itself leads to the release of tumour antigens in the tumour microenvironment. Furthermore, many viruses induce ICD which in turn activates APCs. Given the presence of tumour antigens in the microenvironment, activation and maturation of APCs without additional encoded by virus vector TAAs, probably, may be sufficient to initiate a systemic adaptive immune response to the tumour. The examples of virus vectors applied to stimulate APCs migration, maturation and antigen presentation are listed in Table 2.

**GM-CSF expression.** The first oncolytic viral immunotherapy by HSV expressing the GM-CSF gene (T-VEC) has been approved [169], showing significant tumour control and improved overall survival. In phase III trial OPTiM, in patients with unresectable melanoma, i.t. administrated T-VEC proved clear survival benefits with complete responses in 16.9% patients versus 0.7% of the control group received s.c. injections of recombinant GM-CSF, with best results in earlier stage metastatic melanoma of lower initial tumour size [181]. However, the responses to the therapy in patients with visceral metastases still are unsatisfactory. For this reason, T-VEC is currently being evaluated in combination with CPIs (ipilimumab and pembrolizumab), showing improved activity even systemically [169]. In phase II clinical trial with T-VEC and ipilimumab (NCT01740297) the significant improvement was achieved in combination therapy [182]. However, the results of phase III clinical trial (NCT02263508) with T-VEC and pembrolizumab was disappointing as no significant improvement in survival of patients with advanced melanoma was observed.

In clinical trials for advanced solid tumours, an oncolytic chimeric 5/3 adenovirus producing human GM-CSF—ONCOS-102 (Ad5/3-D24-GMCSF) [161,162] and a poxvirus vector WR strain of VV—JX-594 pexastimogene devacirepvec (Pexa-Vec) [32] were used. Clinical benefits for 8 out of 21 patients treated with ONCOS-102 were observed with proven anti-tumour immune responses. However, a significant anti-adenovirus vector immune response was detected. Over 300 patients have been treated by Pexa-Vec in 12 completed and ongoing clinical trials by intravenous infusion and/or intratumoral injection. Phase II data showed statistically significant improved overall survival for advanced liver cancer patients receiving a high dose of Pexa-Vec. Some of the patients survived for a significantly long time taking into account the advanced stage of cancer. However, in the phase III PHOCUS trial (NCT02562755) enrolling 459 patients with advanced hepatocellular carcinoma, treatment with Pexa-Vec followed by sorafenib was inefficient and showed no improvement compared with the control group. Probably, the use of therapeutic virus is insufficient for treatment of late stages of cancer, which should be addressed in future trials. Further studies are needed to understand which factors predict long-term survival after oncolytic virus treatment [32].

Due to clinically proven efficiency in cancer treatment, many different types of virus vectors were engineered to express GM-CSF alone and in combination with other cytokines. Thus, the anti-tumour efficiency of poxvirus vector—VV Tian Tan strain Guang 9 expressing GM-CSF in different murine tumour models was evaluated alone and in combination with IL-24 [170,171]. Furthermore, the generation of recombinant replication-competent oncolytic mammalian orthoreovirus expressing murine or human GM-CSF (rS1-mmGMCSF and, rS1-hsGMCSF, respectively) was performed [178]. In a murine model of pancreatic cancer rS1-mmGMCSF induced a systemic increase of DCs and promoted T-cell activation after intratumoural administration.

Among Paramyxoviruses, genetically modified NDV MEDI5395 mediated expression of GM-CSF resulted in enhancement of monocyte activation compared with parental NDV [179]. MEDI5395 has broad oncolytic and proinflammatory activity across a range of preclinical human tumour models. In turn, treatment with murine variant NDVmuGM-CSF induced systemic adaptive immunity in CT26 tumours modified to facilitate enhanced viral replication [179].

Alphavirus vector SFV encoding murine GM-CSF (SFV-GM-CSF) was injected intraperitoneally into mice bearing ovarian tumours which resulted in an increase in the number of peritoneal macrophages and neutrophils. Tumour growth was delayed for 2 weeks, but the treatment did not prolong survival [175].

**CD40L expression.** In C57BL/6 mice B16 melanoma model expression of CD-40L by VSV rhabdovirus vector, which selectively replicates in tumours deficient in IFN type I response, showed high levels of T cell activation. However, there was no difference observed in anti-tumour efficacy between the control VSV-GFP and VSV-CD40L. Despite the high T-cell immune response, there was no specificity for TAAs, probably due to high VSV-associated immunogenicity interfering with the priming of tumour-specific T cells, even in the presence of potent co-stimulatory signals. Therefore, VSV, probably, is a poor platform for the priming of prolonged specific T cell responses.

Interestingly, contrary to VSV, intratumoral injection of a replication-defective adenovirus expressing CD40L (Ad-CD40L) resulted in a more efficient anti-tumour response in comparison to both replication-competent VSV-GFP and VSV-CD40L. The Ad-CD40L-mediated tumour regressions were associated with specific T cell responses against TAAs [173]. In phase I/IIa clinical study in 15 melanoma patients, intratumoral injection of replication-deficient AdCD40L expressing CD40L induced desirable systemic immune effects that correlated with prolonged survival [163]. LOAd703 vector, an oncolytic adenovirus with transgenes encoding trimerized membrane-bound CD40L (TMZ-CD40L) and 4-1BBL, has been shown to lyse tumour cells selectively, induce CTL activation and control tumour growth in multiple myeloma xenograft model [164]. Also in the phase I/II trial, patients with unresectable or metastatic PDAC treated with intratumoral injections of LOAd703 and standard intravenous nab-paclitaxel/gemcitabine (nPG) chemotherapy showed an overall response rate of 44%, disease control rate 94% and increase in the proportion of T effector memory cells, while the proportion of Treg and MDSC decreased [165].

**MIP3α expression.** An interesting recombinant oncolytic NDV expressing the macrophage inflammatory protein MIP-3α (NDV-MIP3α) was engineered as an in vivo DC vaccine for amplifying anti-tumour immunity. MIP-3α (CCl20) is a specific chemokine for DCs recruitment. In the NDV-MIP3α treated B16 and CT26 tumour-bearing mice successful attraction of DCs and significant reversion of immunosuppressive tumour microenvironment induced production of tumour-specific cellular and humoral immune responses, which was dependent on CD8^+^ T cells and partially on CD4^+^ T cells [180].

**Fms-like tyrosine kinase 3 ligand (Flt3L) expression**. Fms-like tyrosine kinase 3 ligand (Flt3L) is a growth factor promoting the differentiation and proliferation of DCs. Oncolytic VSV expressing soluble Flt3L was generated to increase the number of DCs and promote tumour antigen presentation in VSV-resistant B16 melanoma and VSV-sensitive E.G7 T lymphoma models [50]. While the combination of VSV and recombinant Flt3L improved animal survival (30% cured animals), the VSV-based expression of Flt3L did not provide a significant treatment advantage in B16 melanoma, and only modest improved survival in the E.G7 model. Furthermore, it was discovered that tumour-associated DCs were actively infected by VSV in vivo, which reduced their viability and migration to the draining lymph nodes to prime a tumour-specific CD8 T cell response. These results demonstrate that VSV inhibits APC functions.

More promising results were obtained with alphavirus SFV vector simultaneously expressing a soluble Flt3L and an XCL1 chemoattractant for classical DC1 (cDC1) cells [177]. Repeated intratumoral injection of the vector led to the delayed progression of syngeneic murine colon cancer MC38 and B16-OVA tumours. The treatment increased the infiltration of CD8+ T cells and facilitated the anti-tumour activity of BATF3-dependent cDC1 in tumour-bearing mice. Furthermore, the SFV therapeutic activity was potentiated by combination with anti–PD-1, anti-CD137, or anti-CTLA-4 immunomodulating antibodies.

### 7.2. Reprogramming of Tumour-Associated Macrophages

High infiltration of immune-stimulating M1 phenotype macrophages in the TME is linked to a Th1-dominant anti-tumour immune response [183]. On the other hand, a high M2/M1 ratio is associated with disease progression. A promising cancer immunotherapy strategy covers the reprogramming of macrophages to the M1 phenotype [184]. Pro-inflammatory cytokines including TNF, IL-12, and IFN can be used to achieve an M1-like phenotype [185,186,187]. Recombinant virus vectors expressing inflammatory cytokines, stimulating phagocytosis and potentially stimulating macrophage transition to the M1 phenotype are listed in Table 2.

**IFN****γ****expression.** IFNγ is known as a potent anti-tumour agent, but its clinical application is limited by its short half-life and significant toxic side effects. The use of viral vectors can help to solve the problem by local intratumoral expression of IFNs directly in TME. Replication-defective adenovirus encoding human IFNγ (Ad-IFNγ) was evaluated in nasopharyngeal carcinoma (NPC) cell lines in vitro and in the xenograft model [166]. The Ad-IFNγ infection resulted in anti-proliferative effects on NPC cells by inducing G1 phase arrest and cell apoptosis. Intratumoral administration of Ad-IFNγ significantly inhibited the growth of CNE-2 and C666-1 cell xenografts in nude mice, while no significant toxicity was observed.

VSV encoding IFNγ (VSVΔ51VSVΔ51-IFNγ) was used for the treatment of 4T1 mammary adenocarcinoma as well as CT26 colorectal murine tumours [174]. The virus vector demonstrated greater activation of DCs and secretion of pro-inflammatory cytokines compared to the parental virus. Tumour growth was suppressed, lung tumours diminished, and prolonged survival was observed in several murine tumour models. The improved efficacy was lost in immunodeficient nude mice proving the T-cell-mediated action of viral therapy.

In the 4T1 mammary carcinoma murine model, alphavirus-mediated delivery of the IFNγ gene to the tumour significantly reduced tumour growth and induced CTL-mediated anti-tumour immune response [176]. SFV-IFNγ expression decreased myeloid cell infiltration into the tumour, and led to an increased number of CD4^+^ and CD8^+^ and a decrease in Treg cell populations [176].

**IFN****α****/****β****expression.** The novel oncolytic poxvirus JX-795 vector (B18R deletion VV mutant) with cloned IFNβ gene (TK-/B18R-/IFN-beta+) demonstrated IFN-dependent cancer selectivity and efficacy in vitro, and tumour targeting and efficacy in mouse models in vivo [37]. In murine adenocarcinoma tumours (CMT-93 and JC) both tumour cells and tumour-associated vascular endothelial cells were targeted after systemic intravenous delivery. The use of high dose intratumoral injection of JX-795 vector (1 × 10^8^ PFU) resulted in 100% complete tumour responses. However, intravenous injection of the same dose was less efficient and induced complete responses in a small number of the treated mice. Local delivery was accompanied by immune-mediated protection against tumour re-challenge.

An oncolytic VSV encoding IFNβ and the sodium iodide symporter (NIS reporter gene, allowing in vivo PET imaging) were designed [49]. Syngeneic acute myeloid leukaemia (AML) C1498 tumours responded to intravenous therapy with VSV-murine IFNβ (mIFNβ)-NIS in a dose-dependent manner. VSV-mIFNβ-NIS was active against a subcutaneous and disseminated murine model of AML as well as AML samples harvested from peripheral blood or the bone marrow of patients. Imaging for NIS expression showed efficient virus infection within the tumours. Virus infection did not increase PD-L1 on tumour cells. Combining VSV-mIFNβ-NIS with anti-PD-L1 antibody therapy enhanced anti-tumour activity compared with treatment with virus alone or antibody alone. Combination therapy with anti-PD-L1 antibody boosted VSV activity with no drug-related toxicities (VSV or anti-PD-L1).

The use of an adenovirus vector expressing IFN α/β (OAd-hamIFN) showed a significant enhancement of the chemoradiation effect in a hamster model of PDAC [167]. IFN α/β expression acted synergistically also with chemotherapy (5-FU, Gemcitabine, and Cisplatin) significantly improving cytotoxic effect in vitro and demonstrating tumour growth inhibition and enhanced survival in vivo.

Finally, replication-deficient adenovirus vector-rAd-IFNα/Syn3, expressing IFNα-2b gene which was combined with a polyamid surfactant (Syn3) enhancing the viral transduction of the urothelium [188], entered a phase III study (NCT02773849) for treatment of non-muscle-invasive bladder cancer [168]. After intravesical treatment more than half of patients with carcinoma in situ achieved complete response during 3 months and half of them were free from high-grade recurrence at 12 months. Clinically meaningful responses were achieved also in patients with high-grade Ta or T1 bladder cancer. Currently, the phase III clinical trial is ongoing, overall providing promising results and probably a novel cancer therapy drug based on adenovirus vector will be approved in near future.

**Anti SIRPα antibody expression.** In order to disrupt the tumour cell CD47 interaction with SIRPα of macrophages, oncolytic poxvirus VV expressing a chimeric molecule that consists of the ectodomain of SIRPα and the Fc domain of IgG4 (SIRPα-Fc-VV) was engineered [172]. The engineered vector SIRPα-Fc-VV replicated successfully in tumour cells, induced the killing of tumour cells in vitro by M1 and M2 macrophages after the addition of conditioned infected cell media to co-cultures, and recruited macrophages and monocytes to tumours in vivo. This effect was not observed for control VVs which either encoded YFP (YFP-VV) or SIRPα (SIRPα-VV). In vivo, in an immune-competent murine F420 osteosarcoma model SIRPα-Fc-VV had greater anti-tumour activity than YFP-VV and SIRPα-VV resulting in significant animal survival. Thus, oncolytic viruses producing SIRPα-Fc may present a promising strategy to enhance the antitumor activity for the virotherapy of solid tumours.

### 7.3. Activation of T and NK Cells

One of the most explored and efficient TME reprogramming strategies is based on the stimulation of the cytotoxic activity of effector T cells and NK cells using Th1 inflammatory cytokines, T cell growth factors, and attraction chemokines. IL-12 has long been known for its potent anti-tumour properties in different animal tumour models, but the clinical experience was disappointing due to severe systemic toxicity and the inability to rich effective concentration in the tumour. Numerous preclinical and clinical studies were performed with different virus vectors, last year’s mostly as combination therapy with other cytokines and immune regulatory molecules. Here presented some examples of recent studies using different vector platforms. The relatively new strategies are (i) the reversion of T-cell exhaustion due to chronic antigenic stimulation using CPIs, and (ii) T-cell anergy reversion by expression of relevant co-stimulatory molecules. Some examples of such strategies applied in research and clinic are described here and listed in Table 3.

**IL-2 expression.** Recent preclinical studies of IL-2 encoding viruses showed promising results. The anti-tumour efficacy of adenovirus coding for an IL-2 variant (vIL-2) protein—Ad5/3-E2F-d24-vIL2 was evaluated in immunocompetent hamsters bearing pancreatic tumours [189]. The expressed vIL-2 was a modified version of IL-2 with a changed binding affinity to subunit β (CD122) of the IL-2 receptor, resulting in decreased recognition by Tregs. Ad5/3-E2F-d24-vIL2 treatment elicited an efficient anti-tumour response achieving a complete response in 62.5% of mice. Moreover, the proportion of immunosuppressive myeloid cells decreased, whereas the tumour-infiltrating CTLs increased as was shown by gene expression analysis of tumours.

Also NDV, the Anhinga strain expressing IL-2 (NDV/Anh-IL-2), effectively inhibited the tumour growth in murine H22 hepatocellular carcinoma [208]. 60 days post-treatment, mice which showed complete tumour regression was well protected against re-challenge with the same tumour cells, confirming the formation of the systemic adaptive immune response.

**IL-12 expression.** In the phase I clinical study, a replication-competent adenovirus (Ad5-yCD/mutTKSR39rep-hIL-12) encoding yCD/mutTKSR39 (yeast cytidine deaminase/mutant S39R HSV-1 thymidine kinase) and human IL-12 was injected intratumorally to 12 patients with metastatic pancreatic cancer at escalating doses [190]. Elevated IL-12, IFNγ, and CXCL10 serum levels were detected in 42%, 75%, and 92% of patients, respectively. The therapy induced immune activation and improved survival.

Next-generation fully virulent oncolytic HSV that does not carry deletions/mutations, but can target HER2 positive (human epidermal growth factor receptor 2) tumours, was developed [194]. Susceptibility to that retargeted virus infection and replication was studied on human breast cancer SK-OV-3 cells and anti-tumour activity at Lewis lung carcinoma murine cell line expressing hHER2 (HER2-LLC1) subcutaneous (s.c.) tumours. The safety profile was very high. Both the parental vector- R-113 and the IL-12-encoding vector- R-115 inhibited the growth of the primary HER2-LLC1 tumour, R-115 being constantly more efficacious. The long-term survivor mice were protected from a second contralateral tumour challenge, providing additional evidence for systemic adaptive immune response development in this model. Analysis of the local response showed that particularly R-115 reversed the immunosuppressive TME, inducing immunomodulatory cytokines, including IFNγ and promoting Th1 polarization. Some of the tumour infiltrating cells, e.g., CD4^+^, CD335^+^ cells, were increased in the tumours of all responding mice, whereas CD8^+^, CD141^+^ were increased and CD11b^+^ cells were decreased preferentially in R-115-treated mice. The durable response included a breakage of tolerance towards both HER2 and the wt tumour cells.

Another second-generation oncolytic HSV encoding murine IL-12 (M002) in metastatic murine ovarian adenocarcinoma led to reduced peritoneal metastasis and improved survival after intraperitoneal injection [195]. Flow cytometry analysis showed the tumour antigen-specific CD8^+^ T cell response in the omentum and peritoneal cavity. The biodistribution and toxicity studies of M032 oncolytic HSV that selectively replicates in tumour cells and encodes for human IL-12 were conducted by intracerebral injection of M032 HSV into *A. nancymae* monkeys [196]. The protocol for phase I clinical study in patients with recurrent or progressive malignant glioma was designed.

WR strain of vaccinia virus VSC20 was engineered to express membrane-bound IL-12 (vvDD-IL-12-FG), as well as secreted IL-12 variant (vvDD-IL-12) [199]. C57BL/6 mice bearing 5-day-old peritoneal murine colon cancer (MC38-luc) were treated by intraperitoneal injection of these virus vectors. Survival results demonstrated that vvDD-IL-12-FG and vvDD-IL-12 elicited potent anti-tumoural effects compared with PBS or vvDD treatment. All the mice that received treatment vvDD-IL-12-FG showed complete regression of the tumours, moreover, subcutaneous re-challenge of MC38 cells did not lead to tumour growth, proving that a systemic tumour-specific immunity was elicited. Furthermore, vvDD-IL-12-FG synergised with anti-PD-1 antibody therapy, leading to the cure of all late-stage MC38 tumours. The immune cell profile analysis in the TME showed a specific increase of CD4+Foxp3− and CD8+ T cells, and the IFNγ secretion from both CD8^+^ and CD4^+^ T cells. Moreover, MDSCs in tumours after IL-12 encoding virus treatment decreased in contrast to parental virus vector. Also, Tregs (CD4^+^Foxp3^+^) were examined and found to be decreased after IL-12-armed virus treatment. A significant decrease was detected in the expression of pro-cancer factors, including TGF-β, COX-2, and angiogenesis markers (VEGF and CD105) after IL-12 encoding virus treatment compared with other treatments. These data demonstrated that vvDD-IL-12-FG treatment, as well as vvDD-IL-12 treatment, successfully turned “cold” tumours into “hot” tumours, which resulted in the extended survival of mice receiving IL-12-expressing virus treatment.

Multiple intratumoral injections of recombinant IL-12 expressing VSV vector (rVSV-IL12) in a murine orthotopic type of mouth squamous cell carcinoma (SCC) model in immunocompetent C3H/HeJ mice caused a significant reduction in tumour volume and substantial survival benefit when compared with saline injections or analogous fusogenic virus (rVSV-F) [205].

The therapeutic efficacy of the novel Measles Schwarz vaccine strain vectors encoding IL-12 fusion protein (MeVac FmIL-12) and an antibody against PD-L1 (MeVac anti-PD-L1), was evaluated in the immunocompetent MC38cea tumour model [209]. Treatment of established tumours with MeVac FmIL-12 achieved 90% complete regressions. Profiling of the TME revealed activation of Th1 immune response, with potent early NK and effector T cell activation as well as upregulation of the IFNγ and TNFα. Additionally, MeVac vectors encoding GM-CSF, the chemokine IP-10 (CXCL10) to activate and recruit effector cells, and antibodies against CTLA-4 and PD-L1 were evaluated. It was found that MeVac encoding the IL-12 fusion protein (FmIL-12) and anti-PD-L1 were the most effective vectors in the murine MC38cea colon carcinoma tumour model.

SFV alphavirus vectors expressing different levels of IL-12 (SFV-IL-12 and SFV-enh-IL-12) were examined [210]. A single intratumoral injection of SFV-IL-12 or SFV-enh-IL-12 vectors into MC38 colon adenocarcinoma induced ≥80% complete tumour regressions with long-term tumour-free survival. The efficiency correlated with IL-12 expression level, as shown by high anti-tumour effects after repeated intratumoral injections of lower doses of SFV-enhIL-12 (carrying translational enhancer) in comparison to SFV-IL-12 vector, providing a lower expression level of IL-12. In all cases, SFV vectors were more efficient in comparison to a first-generation adenovirus vector expressing IL-12 [210]. Moreover, the therapeutic effect of SFV vectors was only moderately affected by animal pre-immunization with SFV. The anti-tumour activity of SFV vectors at least partially, was due to a potent CTL-mediated immune response.

**CD80 (B7.1) expression.** A standard two-dose-escalation phase I clinical trial was conducted with 12 metastatic melanoma patients, using a recombinant VV expressing B7.1 (rV-B7.1) [200]. A partial response was observed in one patient and disease stabilization in two patients. Local immunity was evaluated by quantitative real-time RT-PCR, which revealed that tumour regression was associated with increased expression of CD8 and IFNγ.

Another randomized phase I clinical trial of two recombinant fowlpox viruses encoding human B7.1 or multiple co-stimulatory molecules TRICOM (B7.1, ICAM-1, and LFA-3) was conducted [201]. Fowlpox virus is a replication-deficient and not oncolytic vector shown to induce cell-mediated immunity in animal tumour models. Twelve patients (10 with melanoma and 2 with colon adenocarcinoma) enrolled in the trial were randomized to rF-B7.1 or rF-TRICOM with local administration every 4 weeks. Treatment was well tolerated in patients with metastatic cancer; all subjects exhibited anti-viral antibody responses, but limited tumour-specific T cell responses were detected.

**IL-12, Il-18, and CD-80 (B7.1) co-expression.** The synergistic effects of IL-12, IL-18 and co-stimulatory molecule CD80 have been explored by engineered HSV-1 G47Δ vector [197]. Simultaneously created four oncolytic HSV-1 vectors, which express murine soluble B7.1 (vHsv-B7.1-Ig), murine IL-12 (vHsv-IL-12), murine IL-18 (vHsv-IL-18), and no transgene, were tested in A/J mice harbouring s.c. tumours of syngeneic and poorly immunogenic Neuro2a neuroblastoma. vHsv-IL12 and vHsv-IL-18 demonstrated a stronger anti-tumour effect than any other combinations of two vectors. The triple combination of vHsv-B7.1-Ig, vHsv-IL-12, and vHsv-IL-18 showed the highest efficacy compared to all single vHsv or combinations of two viruses. However, a combination with vHsv-B7.1-Ig did not significantly enhance the efficacy of vHsv-IL-12 at the doses tested. Studies using nude mice indicated that this enhancement of anti-tumour efficacy was likely mediated by T-cell immune responses.

**IL-15, IL-18, and IL-21 (co-)expression.** Oncolytic adenovirus co-expressing IL-12 and IL-18 (RdB/IL-12/IL-18) was engineered [192]. IL-18 stimulates cytotoxicity of NK cells and proliferation of T cells and acts synergistically with IL-12. The effect was investigated by intratumoral administration in the B16-F10 murine melanoma model. The RdB/IL-12/IL-18 therapy improved anti-tumour effects, as well as increased survival. Moreover, the ratio of Th1/2 cytokines as well as the levels of IL-12, IL-18, IFNγ and GM-CSF was markedly elevated in RdB/IL-12/IL-18-treated tumours. Severe necrosis and infiltration of NK cells, as well as CD4^+^ and CD8^+^ T cells, were observed in RdB/IL-12/IL-18-treated tumour tissues.

Another replication-competent adenoviral vector encoding IL-18 (ZD55-IL-18) was used for human melanoma A375 cells and nude mouse A375 tumour xenografts treatment alone or together with melanoma-approved drug—dacarbazine (DTIC) [191]. ZD55-IL-18 and DTIC synergistically inhibited the growth and promoted the apoptosis of A375 xenografts, inhibited VEGF expression, and lung metastasis in xenografts of nude mice.

The use of VSV virus vector encoding IL-15 (VSV-IL-15) in the murine CT-26 tumour model led to the enhancement of anti-tumour T-cell responses and enhanced survival [206]. In another study, using an experimental mouse model of PDAC (Panc02), the natural killer T (NKT) cell activation therapy by α-GalCer-loaded DCs was investigated in combination with a VSV-IL-15 [207]. Panc02 cells were implanted subcutaneously or orthotopically into syngeneic C57BL/6 mice. The anti-tumour effect was compared with single controls: VSV (VSV-GFP), VSV-IL-15 and NKT cell therapy (α-GalCer-loaded DCs). Superior tumour regression and increase in survival were observed for combined treatment with VSV-IL-15 and NKT cell therapy. Furthermore, the addition of PD-1 blockade significantly improved monotherapy with oncolytic VSV-IL-15 representing a promising treatment strategy for pancreatic cancer.

Engineered Lister strain of VV with additional deletion of the VV 13.8-kDa N1L protein, a neurovirulence factor (named VVLΔTKΔN1L), and encoding IL-21 (VVLΔTKΔN1L-mIL-21) enhanced anti-tumour immune responses in murine models of colorectal cancer. The adaptive T cell responses able to eliminate primary tumours and induce the development of systemic anti-tumour immunity preventing tumour recurrence were demonstrated [202].

**Antibodies to PDL1 expression.** A high-capacity inducible adenoviral vector (HCA-EFZP-aPDL1) for the controlled expression of PD-L1 blocking antibody was engineered [193]. The vector was tested in an immune-competent mouse model of colorectal cancer (MC38). A single local administration of HCA-EFZP-aPDL1 in s.c. lesions led to a significant reduction of tumour growth with minimal release of the antibody in the circulation. However, in the rapidly progressing peritoneal carcinomatosis model, the anti-tumour effect was weak even in combination with other immune-stimulatory agents. Macrophage depletion enhanced the efficacy of HCA-EFZP-aPDL1, suggesting the importance of inflamed TME induction for efficient CPI immunotherapy.

A promising class of therapeutic antibodies is recombinant antibodies of dual specificity, also called bispecific T cell engagers (BiTEs), comprised of two single-chain variable fragments (scFvs) that recognize a target TAA along with T cell receptor (usually CD3ε) [212]. This type of antibody simultaneously binds to CTL and to tumour cells via the TAA part resulting in MHC-dependent antigen presentation and activation of CTLs. BiTEs have shown promise as anti-tumour therapeutics, as approved by the FDA blinatumomab for treating acute lymphoblastic leukaemia, which engages the CTL to CD19^+^ tumour. To target T cell cytotoxicity directly towards PD-L1-expressing cells, a BiTE crosslinking PD-L1 and T cell receptor CD3ε (anti-PD-L1 scFv plus anti-CD3 scFv) was developed on the basis of oHSV-1 G207 vector backbone (oHSV-1 PD-L1 BiTE) [198]. Successful delivery and targeted cytotoxicity were demonstrated in the patient-derived ascites model. This approach activated endogenous T cells within malignant ascites, generated a pro-inflammatory response, and eliminated PD-L1-positive tumour cells and macrophages, while leaving T cells unaffected. The use of a virus vector for local expression of PD-L1 BiTE also prevented systemic toxicities.

CCF33-hNIS-anti-PD-L1 is an oncolytic poxvirus encoding two transgenes: human sodium iodide symporter (hNIS) and a single-chain variable fragment scFv ab against PD-L1 [203]. The phase I clinical trial is planned for patients with triple-negative breast cancer (TNBC). In a xenograft model of TNBC (MDA-MB-468), CF33-hNIS-anti-PD-L1 was able to completely control tumour growth already at a low dose. Comprehensive preclinical pharmacology studies were performed to support the clinical development of CF33-hNIS-anti-PD-L1 [203].

SFV and AAV vectors expressing anti-PDL1 (aPDL1) monoclonal antibody were engineered [211]. The SFV-aPDL1 induced >40% complete regressions and was superior to AAV-aPDL1 treatment, as well as to monotherapy with anti-PDL1 given systemically or locally in a murine MC38 tumour model. The higher SFV-aPDL1 anti-tumour activity could be related to the interferon response induced by SFV RNA replication.

**CCL5 (RANTES) expression.** An oncolytic VV expressing CCL5 (vvCCL5), induced chemotaxis of lymphocyte populations in vitro and in vivo, and displayed immunotherapy proved safety in vivo [204]. The vvCCL5 showed significant tumour suppression and enhanced survival compared to vvDD mock-treated C57/BL mice bearing s.c. M38 tumours. Interestingly, enhanced therapeutic benefits with vvCCL5 in vivo correlated with increased persistence of the virus vector within the tumour. Vaccination of tumour-bearing mice with DCs further improved vvCCL5 anti-tumour efficiency which correlated with increased levels of tumour infiltrating lymphocytes.

### 7.4. Targeting Extracellular Matrix and Vasculature

A very important supportive strategy to enhance tumour infiltration by Th1 type immune cells and promote therapeutic virus, or drug spread is to interfere with TGFβ signalling and reduce the ECM density and tumour interstitial pressure. In this respect, here are described a few strategies using adenoviruses (Table 4).

**Targeting TGF****β****signalling.** The oncolytic adenovirus rAd.sT, encoding soluble TGF receptor II fused with human IgG Fc fragment (sTGFβRIIFc) gene under the control of telomerase reverse transcriptase promoter was created [213]. In the immunocompetent mouse 4T1 breast tumour model, intratumoral delivery of sTGFβRIIFc inhibited both tumour growth and lung metastases. Downregulation of the expression of several TGFβ target genes involved in tumour growth and metastases, inhibition of Th2 cytokine expression, and induction of Th1 cytokines, chemokines, as well as granzyme B and perforin expression, were observed. Oncolytic treatment also increased the percentage of CD8^+^ T lymphocytes, promoted the generation of CD4^+^ T memory cells, reduced Tregs, and reduced bone marrow-derived suppressor cells. Importantly, rAd.sT treatment increased the percentage of CD4^+^ T lymphocytes, and promoted differentiation and maturation of antigen-presenting DCs in the spleen.

**Hyaluronidase expression.** To enhance virus spread within tumours a replication-competent adenovirus expressing a soluble form of the human sperm hyaluronidase (PH20) under the control of the major late promoter (MLP) (AdwtRGD-PH20) was generated [214]. Hyaluronidase is an enzyme which dissociates the ECM and could enhance the intratumoral distribution of therapeutic virus particles and improve its therapeutic activity, especially for tumours highly expressinghyaluronan, such as pancreatic cancer. Treatment of human highly expressing hyaluronan melanoma SkMel-28 nude mouse xenografts with AdwtRGD-PH20 resulted in degradation of hyaluronan, enhanced viral distribution, and tumour regression. Furthermore, the PH20 cDNA was inserted into another oncolytic adenovirus that selectively kills retinoblastoma (Rb) pathway-defective tumour cells. The anti-tumour activity of the novel oncolytic adenovirus expressing PH20 (ICOVIR17) was higher in comparison to that of the initial AdwtRGD-PH20 virus.

VCN-01 is a replication-competent adenovirus with enhanced infectivity through a modified fibre specifically engineered to replicate in tumours with a defective Rb pathway, encoding a soluble hyaluronidase [215]. The anti-tumour effect of VCN-01 was evaluated in vitro in osteosarcoma patient-derived cell lines and in vivo in orthotopic intratibial and lung metastatic osteosarcoma murine models. VCN-01, after both intratumoral and systemic administration, showed a potent anti-sarcoma effect.

### 7.5. Simultaneous Targeting of Multiple Pathways

The main goal and one of the remaining challenges in cancer immunotherapy is the development of a long-lasting specific adaptive immune response against the tumour. It is evident that different compartments of TME have to be targeted simultaneously to achieve this goal. The experiments employed virus vectors expressing multiple cytokines and/or inhibitors of growth factors undoubtedly proved the superior therapeutic efficiency and encouraged using such kind “cocktail” treatments. Some examples of cytokine combinations targeting both APCs and T cells or tumour ECM modifying molecules expressed by virus vectors are described below and summarised in Table 4.

#### 7.5.1. Combined Activation of T cells/NK Cells and DCs/Macrophages

**Expression of IL-2 plus TNF alpha.** With the aim to stimulate immunological “danger” signalling and T-cell trafficking/activation, the adenoviruses expressing TNFα and IL-2 were examined [6]. The virus injections were initiated prior to anti-PD-1 antibody treatment in a prime-boost approach, which led to complete murine B16.OVA melanoma tumour regression with all treated mice being cured. Virus expression of IL-2 and TNFα altered the cytokine balance in the TME towards Th1 and increased the intratumoral proportion of both CD8^+^ and CD4^+^ T cells. This preclinical study provided a background for an ongoing clinical trial with oncolytic adenovirus encoding TNFα and IL-2 (TILT-123) in melanoma patients who receive an anti-PD-treatment [223,224].

**Expression of IL-12 plus CCL2.** HSV-1 vector M002, a γ(1)34.5-deleted HSV-1 that expresses IL-12, was tested in combination with parental HSV-1 engineered to express the macrophage-attracting chemokine CCL2 [220]. Neuroblastoma Neuro-2a tumours were established subcutaneously in the syngeneic A/J mouse strain. The inhibition of tumour growth was demonstrated, which was most efficient in tumours treated with a combination of M002 and M010 vectors.

**Expression of IL-12, IL-15, IL-7, CCL19, PD-1v plus GM-CSF.** The anti-tumour efficacy of HSV-2 vectors engineered to express IL-12, IL-15, PD-1v, GM-CSF, and IL-7 - CCL19 were evaluated in syngeneic CT26 tumour-bearing and 4T1 tumour-bearing mice [221]. It was found that all variants had a similar anti-tumour activity, although the oHSV2-GM-CSF and oHSV2-IL-7 -CCL19 were slightly more efficient than the other three viruses (oHSV2-IL-12, oHSV2-IL-15, oHSV2-PD-1v). Interestingly, in this study oHSV-2 encoding GM-CSF showed better efficacy than the IL-12 encoding vector of the same backbone. The most potent activity was observed for all five virus vector combinations. The tumour re-challenge revealed that the “cocktail” therapy resulted in a systemic lasting anti-tumour immune memory preventing secondary identical tumorigenesis.

**Expression of IL-12 plus GM-CSF.** Retargeted to the human HER2 receptor, not attenuated oncolytic HSV vector encoding murine IL-12 (mIL-12) was modified by insertion of a second immunomodulatory molecule, murine GM-CSF (mGM-CSF), to maximize therapeutic efficacy [222]. This double-armed (R-123) virus was evaluated in HER2-LLC1 tumour-bearing mice. The R-123 vector was compared to singly expressing GM-CSF (R-121) and IL-12 (R-115) oHSVs vectors. While monotherapies with either unarmed and armed retargeted HSVs were only moderately effective, the combined treatment of all variants with anti-PD1 led to a significant improvement in efficacy, and the percentage of complete tumour response comprised 60% (HSV), 75% (mIL-12), 50% (mGM-CSF), and 100% (mIL-12+mGM-CSF) of treated mice. Importantly, systemic delivery of double-armed virus combined with anti-PD1 was effective in inhibiting the development of tumour metastasis. Furthermore, the targeted HSVs armed with multiple cytokines were effective upon local and systemic delivery and were able to cure advanced established tumours.

**Expression of PD-L1 antibody plus GM-CSF.** Oncolytic vaccinia virus vector (VV-iPDL1/GM) was generated to co-express a murine soluble PD-1 extracellular domain fused with IgG1 Fc as a PD-L1 inhibitor (iPDL1) and murine GM-CSF [3]. The antitumor activity of VV-iPDL1/GM was investigated in weakly immunogenic B16-F10 melanoma expressing luciferase reporter. Intratumoral injections with VV-RFP or VV-GM-CSF strongly inhibited tumour growth leading to complete responses. Nevertheless, the effect of VV-iPDL1/GM was more potent. The anti-tumour effect of the recombinant VVs was also significant in Py230 breast cancer and MC38 colon adenocarcinoma murine tumours [3]. In vivo CD8^+^ T cell depletion significantly decreased the systemic anti-tumour activity of the VV-iPDL1/GM vector. Furthermore, the increased DCs (CD11c^+^) content in tumours treated with VV-iPDL1/GM vector was observed as well as reduced MDSCs of the CD11b^+^ population. Moreover, this virus vector induced tumour neoantigen-specific T cell responses demonstrating a promising potential for the treatment of poorly immunogenic tumours.

**Expression of CTLA-4 antibody plus GM-CSF.** Recombinant replication-competent Ad5 vectors SKL001 SKL002 encoding GM-CSF and anti-CTLA-4 antibody, respectively, showed the selective replication and anti-tumour activity in human tumour A549 lung xenograft, murine B16F10 melanoma, and CMT-64 mouse lung carcinoma models [216]. A more potent anti-tumour effect was observed in the case of a combination of both viruses. Combination with GM-CSF encoding virus vector showed enhancement in a number of tumour mature dendritic cells and macrophages in contrast to CTLA-4 virus treatment alone.

#### 7.5.2. Combined Activation of Immune Cells and Stroma

**TGF****β****binding short hairpin RNA plus GM-CSF plus TAA.** Oncolytic adenovirus encoding mouse GM-CSF (AdG) was engineered to express also short hairpin RNA of mouse TGFβ 2 (shmTGF-β2) gene (AdGshT) [217]. Additionally, plasmid DNA expressing MART1, a human melanoma-specific tumour antigen was used as a DNA vaccine. Each virus was intratumorally injected into melanoma-bearing C57BL/6 mice. As a result, mice that received AdGshT showed delayed tumour growth than those that received AdG. Immune activation was mainly induced by mature tumour-infiltrating DCs and decreased Tregs in tumour-infiltrating lymphocyte populations. The combination of tumour antigen-specific induction via MART1 with the non-specific immune stimulation via GM-CSF and shTGF-β2-mediated anti-tumour effects in presence of the oncolytic adenovirus was more potent than the anti-tumour effects of individual treatments (MART1 or GM-CSF/shRNA of TGF-β2). However, complete regression of tumours was not observed. Authors suppose that the immune response to the vector inhibited the spread and persistence of the virus within the tumours and hampered the specific anti-tumour response.

**Relaxin plus GM-CSF and IL-12.** The antitumor efficacy by a combination of oncolytic adenovirus (oAd/IL12/ GM-RLX), which co-expresses relaxin (RLX), IL-12 (IL-12p35 and IL-12p40), and GM-CSF in combination with anti-PD-1 was examined in hamster s.c. and orthotopic pancreatic tumour models [218]. Relaxin-expressing oncolytic Ad—oAd/RLX effectively degraded tumour ECM and enhanced the tumour penetration of trastuzumab in comparison with trastuzumab monotherapy. It was demonstrated that the expression of four genes, mediated by a single oncolytic Ad vector oAd/IL12/GM-RLX can modulate both physical and immunological properties of the TME.

**VEGF binding short hairpin RNA plus IL-12.** To overcome tumour-mediated immunosuppression and enhance the potency of immune gene therapy, oncolytic adenovirus (Ad) co-expressing IL-12 and VEGF-specific short hairpin ribonucleic acid (shVEGF; RdB/IL12/shVEGF) was evaluated [219]. Intratumoral injection of RdB/IL12/shVEGF vector induced a strong anti-tumour effect in an immune-competent B16-F10 melanoma model. Local delivery of RdB/IL12/shVEGF to tumour tissues resulted in massive infiltration of differentiated CD4^+^ T cells, CD8^+^ T cells, NK cells, and DCs. Furthermore, RdB/IL12/shVEGF induced a potent tumour-specific Th1 immune response and efficiently suppressed the expression of VEGF.

## 8. Summary

The approval of some virus vector-mediated therapies for cancer treatment in the USA and Europe as Talimogene laherparepvec (T-VEC) in 2015/2016 for melanoma treatment opened the door for intensive work by introducing cancer virotherapy into the clinic and encouraged researchers to test untrivial treatment strategies targeting specific TME processes. The individual sensitivity to the virus vector infection and replication seems to be crucial for the virotherapy success. The enormous diversity of viruses brings great promise for individually tuned personalized virotherapy. The most adapted and clinically developed are DNA virus vectors, which possess great stability, easy engineering and good capacity for the introduction of multiple therapeutic genes. The limitation for any DNA virus vector in clinical practice is the potential ability of non-specific vector integration and insertional mutagenesis (gene toxicity). Especially, this could be the case for herpesviruses and adenoviruses replicating in the cell nucleus. On the other hand, the ability of herpesviruses to establish latent infection within specific tissues potentially could be valuable in terms of cancer therapy. 

Although non-integrating DNA viruses replicating in the cell nucleus are proposed to be safe in respect of insertional mutagenesis, the risk of non-specific chromosomal integration of viral DNA should not be ignored, considering the genetic instability and heterogeneity of cancer cells, especially in case of repeated local injection of high titre virus preparations. The experimental setting to evaluate the probability of random integration and points of insertions is rather sophisticated and difficult, because of that such works are rare [225]. Nevertheless, such risks were assessed regarding adenovirus vectors showing, that the probability of heterologous integration of adenovirus vector is higher in comparison to homologous one in murine models [226,227]. Furthermore, in earlier studies, it was observed that injection of hamsters with wild-type adenovirus type 12 (Ad12) led to tumour development due to chromosomal integration of virus DNA and expression of the E1A/E1B oncoproteins [228].

Another important problem is a strong adaptive antiviral response and pre-existing vector immunity, characteristic of widely used adenovirus, herpesvirus and poxvirus vectors. The major objectives in adenovirus vector development are to overcome the challenges associated with strong innate immune responses to its capsid proteins, and robust adaptive immune responses to de novo synthesized viral and transgene products. The strong immune response is limiting the efficiency of virus replication and interferes with the immune response to the TAA, which is crucially important for long-term anti-tumour protection. Therefore, to prolong the therapeutic effect of introduced immune-modulation genes and to improve the efficiency, a prime-boost by different vector combinations can provide a solution. On the other hand, the innate response against the intratumorally administered virus vector in presence of relevant memory T- cells due to previous exposure to the parental virus can be sufficient to overcome the TME immunosuppressive conditions and boost the efficient adaptive anti-tumour immune response [229,230]. Therefore, the pre-existing immunity towards the vector may promote the immune cell activation.

Also, cytokine modifications improving their half-life, e.g., expression of membrane-bound cytokines, are perspectives for prolongation of specific cytokine action in the tumour. In terms of efficiency of therapy, the replication-competent virus vectors should be preferred for local intratumoral administration, whereas replication-deficient vectors can be used for systemic delivery, if tumour tropism is expected. Systemic vector delivery is important for metastases treatment. The great possibility for systemic tumour targeting provides VVs, which are naturally resistant to complement and antibody neutralization [34].

The ability of virus vectors to infect also the immune cells is a clearly underestimated option, which has to be taken into account, as it affects the ability to induce long-lasting immune response, the main goal of immunotherapy. Such an example is shown and discussed regarding infection of DCs by VSV vector [50]. In such a case one of the solutions could be VSV vector engineering to diminish the vector’s lytic properties together with direct expression of particular TAA. Also, the ability of VSV for pseudotyping can be used to avoid not desired immune cell infection. Except for VSV, the potential of RNA virus vectors is less exploited for cancer therapy. RNA vectors are very promising, because many RNA vectors do not induce strong adaptive immune responses, allow a high level of transgene expression, and also provide more possibilities for specific tumour tissue targeting. It is clear, that the development of the new RNA vectors and their clinical translation, e.g., vectors based on enteroviruses and arenaviruses, would be highly appreciated.

The intrinsic ability of virus vectors to reverse immunosuppression and provide tumour-specific neoantigens inducing adaptive T cell response, has demonstrated absolutely incredible results in combination with new CPI therapy. Checkpoint inhibition tends to work in “hot” tumours characterized by CTLs infiltration, while little efficacy is seen in “cold” tumours [231]. Recent clinical research clearly showed that the use of CPIs together with virotherapy significantly improves therapeutic efficiency [169]. Furthermore, results of preclinical research combining cytokines and CPIs expressing viruses are impressive, showing complete response nearly in all treated animals with durable immunity [209]. The virus vector-mediated expression of CPIs ensures higher local concentrations in the target tumour TME and avoids undesirable toxicity. The potential limitation of anti-PD-L1 antibodies as well as BiTE-based therapies is the plasticity of tumours, due to antigenic heterogeneity enabling the selection of variants not recognized by the engineered antibody.

Lytic and immunogenic properties of the viruses potentially can synergize with chimeric antigen receptor (CAR)-modified T cells [232]. Viral vectors targeting ECM (hyaluronidase, relaxin) or other factors could enhance intratumoral CAR-T distribution solving the main treatment bottleneck related to CAR-T application in solid tumours. It was shown that oncolytic Ad expressing RANTES and IL-15 [233] or TNFα and IL-2 [234] stimulated intratumoral CAR-T accumulation in combined treatment. Although the changes in immune cell composition cannot be assessed in these studies due to the use of NOD SCID gamma (NSG) mice model, authors hypothesize that killing of target cells may be more efficient due to enhanced delivery of CAR-Ts and synergy with viral vectors. The efficiency of CPIs treatments not only shows the importance of immune stimulatory factors, but also indicates the crucial requirement to remove/eliminate the immune suppressive factors to convert “cold” TME to a “hot” state. Most current immune therapies are focused on stimulation, and much less explore the inhibition of immune suppressive cytokines, such as IL-10. IL-10 is a key immunosuppressive cytokine that impairs proliferation, and cytokine production, promotes MDSCs and inhibits DCs differentiation [235,236]. It was found that IL-10 concentration in cancer patients’ blood correlated with overall survival, as well as the level of TGFβ was also frequently elevated in patients with high IL-10 [163]. We suppose that viruses expressing inhibitors of IL-10 or its soluble receptor would have great potential for TME reprogramming. Additionally, inhibition of TGFβ1 and VEGF together with immune stimulation could be a highly prospective therapeutic strategy. Some initial research, regarding TGFβ inhibitors is ongoing [217].

In general, it is rather difficult to compare the anti-tumour activity of different vectors expressing cytokines, as the tumour models, virus vector backbones, and study design have diverged. Interestingly, in one study it was reported that an oncolytic HSV vector armed with IL-12, showed better efficacy than a GM-CSF armed vector of the same backbone in certain tumour-bearing mice models [237], but in the study of Hu et al., oHSV-2 encoding GM-CSF showed opposite better efficacy than IL-12 armed vector of the same backbone [221]. Currently, there are very few studies which compare different vectors with the same transgenes using the same tumour model. Therefore, the efficiency of vectors for TME programming is difficult to assess. Nevertheless, it seems that a more significant anti-tumour effect was achieved by the use of pro-inflammatory cytokines in combination with anti-PD1/PDL1 antibodies, such as demonstrated for IL-12 [209] and IL-2 with TNFα [6] encoding virus vectors.

## 9. Concluding Remarks

The understanding of general immunosuppressive TME features, the discovery of CPIs and the achievements in tumour selective virus vector engineering with numerous vector choices bring great opportunities for further successful use of virotherapy in cancer treatment. The individual differences in immune cell composition, as well as in response to the tumour immunotherapy during the treatment, should be studied carefully. Different treatment strategies should be developed and applied depending on the inflamed or poorly immunogenic tumour status. The establishment of individual tumour/virus vector sensitivity and TME status assessment before the therapy would be of great priority. Probably, biopsy analysis for the virus receptor expression, PD1/PD1L expression, immune cell composition, as well as for such cytokines as TGFβ and IL-10 would help to establish efficient personalized cancer immunotherapy with available “off-the-shelf” virus vectors. The use of a reporter system such as NIS allowing for tracking virus replication during the treatment by positron emission tomography also would help in the assessment of the correct therapeutic virus choice. For the viruses with a high probability of pre-existing immunity, such as adenoviruses and MV, the antibody levels should be checked before the therapy and during it. Future directions should cover the rules for the selection of a personalised virus treatment approach, based on (i) tumour sensitivity to the chosen virus vectors, (ii) TME composition evaluation, and (iii) monitoring virus vector replication and cytokine release during patient treatment. Viral vectors offer broad opportunities for efficient, safe and non-toxic cancer immunotherapy and definitely would become a standard tool in the cancer treatment arsenal.

## Figures and Tables

**Figure 1 biomedicines-10-02142-f001:**
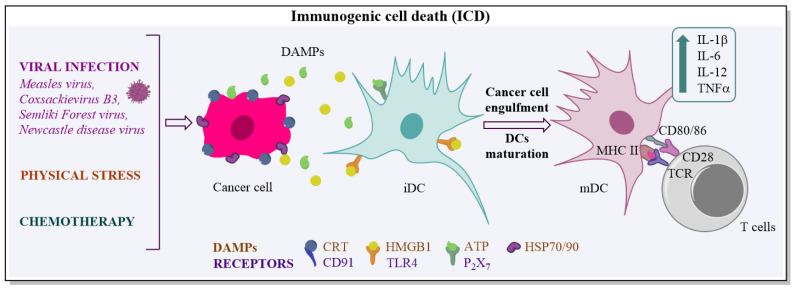
Activation of the adaptive immune response through immunogenic cell death (ICD). ICD can be induced by viruses, and physical and chemical stress. In the process of ICD cancer cells release danger-associated molecular patterns (DAMPs) including ATP, nuclear high mobility group box 1 (HMBG1), calreticulin (CRT) and heat shock proteins (HSP) 70 and 90. These signals activate dendritic cells (DCs) and promote their maturation, secretion of inflammatory cytokines and tumour-associated antigen presentation to T cells. iDC—immature DC, mDC—mature DC, TCR—T cell receptor, TNFα—tumour necrosis factor α, TLR—Toll-like receptors.

**Figure 2 biomedicines-10-02142-f002:**
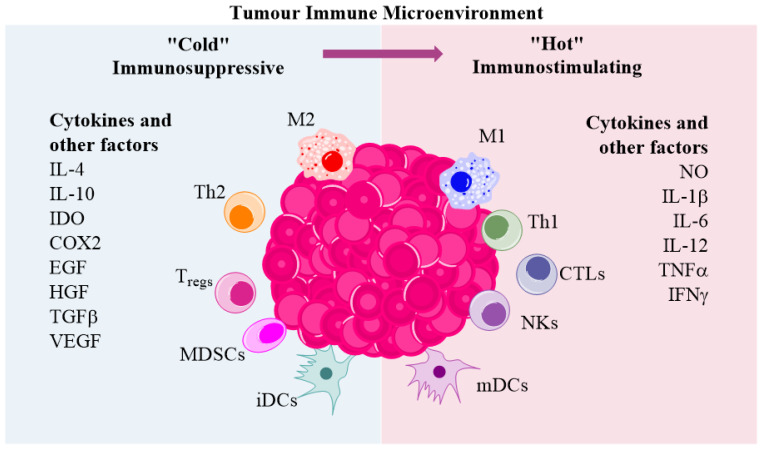
Composition of the “Cold” immunosuppressive tumour microenvironment versus “Hot” immunostimulating tumour microenvironment (TME): immune cells, cytokines and other factors. TME reprogramming towards a “hot” state is a promising cancer therapy strategy. M1—classically activated macrophages, M2—alternatively activated macrophages, Th1—type I T helper cells, Th2—type II T helper cells, Tregs—regulatory T cells, CTLs—cytotoxic T lymphocytes, NKs—natural killer cells, MDSCs—myeloid-derived suppressor cells, iDCs—immature dendritic cells, mDCs—mature dendritic cells. IDO—indoleamine-2,3-dioxygenase, COX2—cyclooxygenase-2, NO—nitric oxide.

**Figure 3 biomedicines-10-02142-f003:**
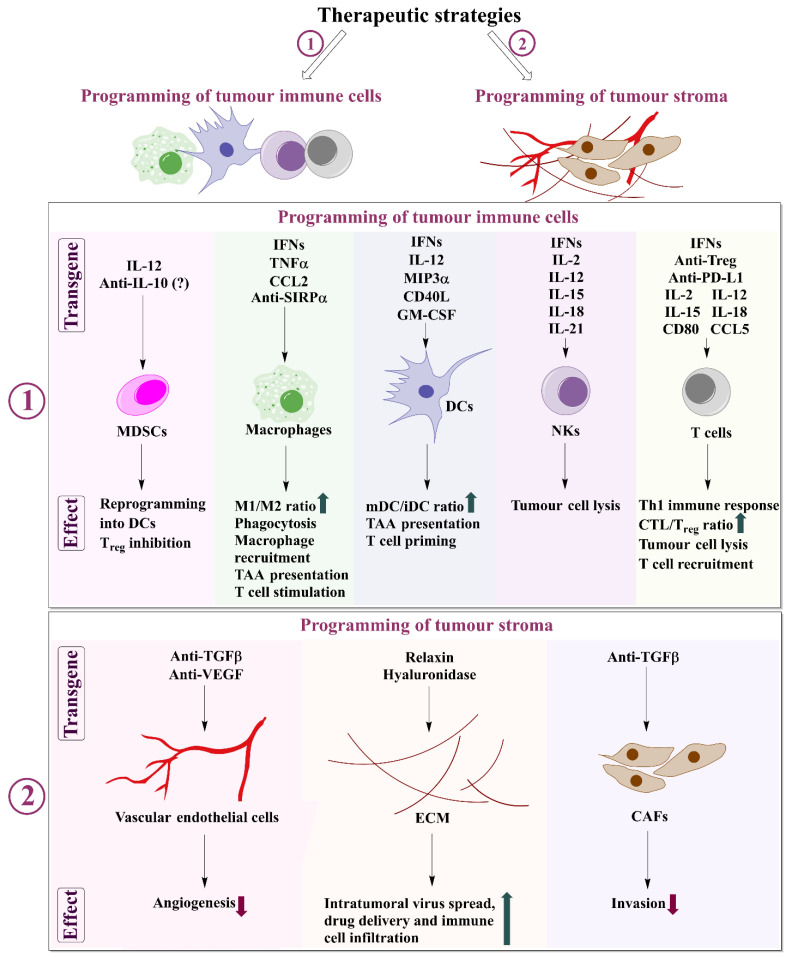
Cancer therapy strategies for programming tumour immune cells ① and tumour stroma ②. Transgenes expressed by viral vectors promote a “hot” immunostimulating tumour microenvironment and support anti-tumour immunity (effects). Green arrows represent upregulation of the effect, while red arrows – downregulation. M1—classically activated macrophages, M2—alternatively activated macrophages, CAA—cancer-associated antigens, Th1—type I T helper cell, Treg—regulatory T cell, CTL—cytotoxic T lymphocyte, NKs—natural killer cells, MDSCs—myeloid-derived suppressor cells, iDC—immature dendritic cell, mDC—mature dendritic cell, DC—dendritic cells, ECM—extracellular matrix, CAFs—cancer-associated fibroblasts.

**Table 1 biomedicines-10-02142-t001:** Comparison of viral vectors.

Virus Type	Insert Capacity	Cell Receptor/Tropism	Advantages	Limitations
DNA vectors				
Adenoviruses	up to 7.5 kb up to 36 kb (fully deleted helper dependent Ads)	Coxsackie Adenovirus receptor (CAR); CD46	high transduction efficiency broad tissue tropism availability of scalable production system tumour-specific gene promoters	pre-existing viral immunity strong immune responses against vector proteins biosafety concerns (random integration)
Poxviruses	up to 24 kb 7.5 kb (MVA)	binding to glycosaminoglycans following cell fusion; virus replication and spread are dependent on epidermal growth factor receptor (EGFR) signalling; preferential replication in cancer cells	inherently tumour targeting cytoplasmic replication low prevalence of anti-vector immunity large production of clinical grade preparations available stable in blood following intravenous injection and highly efficient systemic delivery	replication-deficient Poxvirus vectors encoding heterologous antigens have a lower ability to prime immune responses in humans than other viral vectors adaptive immune response against the vector large virus particles hampering their intratumoral spread
Herpesviruses	up to 40 kb (replication-deficient vector) up to 14 kb (HSV1)	Herpesvirus Entry Mediator and nectin 1 (HSV-1)	selective replication in tumours potent cytolytic capability blood-brain barrier crossing availability of scalable production system	potential neurovirulence (HSV-related encephalitis) genetically modified HSV vectors are not very efficient compared to oncolytic wild-type variants pre-existing immune response strong immune responses against vector proteins
RNA vectors				
Rhabdoviruses	4–6 kb	multiple receptors were proposed (phospholipids and gangliosides, nicotinic acetylcholine receptor, neural cell adhesion molecule, and low-density lipoprotein gene family receptors (LDLR)	selective and efficient replication in tumour cells including in metastases high oncolytic properties pseudotyping capabilities ability to cross the blood-brain barrier (VSV)	infection of tumour-associated DCs reduces their antigen presentation properties potential neurovirulence (VSV) insufficiently developed large-scale manufacturing technology
Alphaviruses	up to 5 kb	very low-density lipoprotein receptor (VLDL-R) and apolipoprotein E receptor 2 (ApoER2)	low specific immune response against the vector low pre-immunity tumour tropism (SIN) induction of immunogenic cell death high level of transgene expression	modest insert capacity short time expression potential neurovirulence (SFV) insufficiently developed large-scale vector production system
Arenaviruses	up to 2 kb	preferentially infect monocytes, macrophages, and DCs through binding to α-dystroglycan (α-DG)	non-lytic infection of dendritic cells efficient CD8^+^ T cell immunity weak neutralizing antibody response against the vector rare pre-existing anti-vector immunity safe in human	low insert capacity limited direct oncolytic properties insufficiently developed large-scale vector production system
Enteroviruses	0.3–1.7 kb	immunoglobulin-like receptor, CD155; Nectin-like molecule 5 (Poliovirus), coxsackie-adenovirus receptor (CAR); RGD motif of integrins (Coxsackievirus), other co-receptors	tumour tropism neurotropism (Poliovirus) low pathogenicity oncolytic replication	very low insert capacity unstable genome high levels of pre-existing immunity to polio vectors insufficiently developed large-scale vector production system
Reoviruses	up to 1.5 kb within two RNA segments	the receptor is unknown, but is thought to include sialic acid and junctional adhesion molecules (JAMs)	oncolytic properties relatively non-pathogenic in adults tumour tropism	insufficiently developed recombinant vector platform transient expression anti-vector pre-immunity (neutralising Abs) very low insert capacity
Paramyxoviruses	up to 6 kb (Measles virus, MV) 4.5 kb (Newcastle disease virus, NDV)	different receptors: *MV*: signal lymphocyte-activation molecule (SLAM or CD150) CD46, Nectin-4 *NDV*: sialic acids on the tumour cell surface	tumour tropism pDC maturation oncolytic properties low seroprevalence (NDV) safe for human	pre-existing immunity (MV) insufficiently developed recombinant vector platform

**Table 2 biomedicines-10-02142-t002:** Virus vectors encoding cytokines and other molecules activating DCs and macrophages.

Transgene	Virus Vector (Virus Backbone)	Model	Effect	Ref.
**Adenoviruses**				
GM-CSF	ONCOS-102 (Ad5/3-D24-GMCSF)	patients advanced solid tumours	biodistribution and toxicity study in Syrian hamster showed broad tropism. The virus DNA expression was detectable nearly for one month. No severe adverse events occurred in 21 patients with advanced solid tumours. Clinical benefits for 8 out of 21 patients with confirmed anti-tumour immune responses; however significant anti-adenovirus vector immune response was also detected.	[161,162]
CD40L	AdCD40L (Ad5)	melanoma patients (*n* = 15) phase I/IIa study	induced desirable systemic immune effects that correlated with prolonged survival	[163]
TMZ-CD40L plus 4-1BBL	LOAd703 (Ad5/35)	panel of human multiple myeloma cell lines (ANBL-6, L363, LP-1, OPM-2, RPMI-8226, and U266-84), RPMI-8226 xenografts patients with late-stage pancreatic cancer (PDAC)	in preclinical multiple myeloma studies: selective tumour cell lysis, induction of CTL activation, control of tumour growth; in phase I/IIa clinical study overall response rate of 44%, disease control rate 94% and increase in the proportion of T effector memory cells, while the proportion of Treg and MDSC decreased	[164,165]
IFNγ	Ad-IFNγ (Ad5)	murine nasopharyngeal carcinoma (NPC), CNE-2 and C666-1 cell xenografts in nude mice	anti-proliferative effects in NPC cells; xenograft tumour growth inhibition in nude mice	[166]
IFNα/β	OAd-hamIFN (Ad5)	pancreatic cancer (PDAC), hamster	IFN expressed from OAd-hamIFN acts synergistically with radiation and chemotherapy significantly improving cytotoxic effect in vitro and inhibiting tumour growth in vivo, resulting in prolonged survival	[167]
IFNα-2b	rAd-IFNα/Syn3 (Ad5)	phase III clinical study, patients with non-muscle-invasive bladder cancer	53.4% (55 patients) of 103 patients with carcinoma in situ had a complete response within 3 months and it maintained in 25 (45.5%) of 55 patients at 12 months	[168]
**Herpesviruses**				
GM-CSF	T-VEC (ΔICP34.5 oHSV1)	clinical trials with metastatic stage IIIB/C–IVM1a melanoma	proved significant systemic disease control, especially in combination with antiCTLA-4 (ipilimumab), and antiPD1 (pembrolizumab); infiltration of TAA-specific CD8^+^ and CD4^+^ T-cells and inflammation in tumours, decrease in MDSC and Treg populations	[169]
**Poxviruses**				
GM-CSF	PexaVec—JX-594 (VV-WR strain)	Patients with renal cell cancer, colorectal cancer, hepatocellular carcinoma	in patients with advanced HCC overall survival was significantly longer in the high-dose arm (median 14.1 months versus 6.7 months at low dose); induction of dendritic cell maturation and increase in leukocytes numbers in patients’ blood	[32]
GM-CSF	VG9-GMCSF (VV Tian Tan strain Guang 9)	murine melanoma B16, s.c. tumour	significant inhibition of tumour growth, prolonged survival and cytotoxic immune response	[170]
GM-CSF plus IL-24	VG9-GMCSF-IL24 (VV Tian Tan strain Guang 9)	murine cancer cell lines B16, 4T1, MDA-MB-231, CT26, HCT116, A549; B16, 4T1 and CT-26 sc murine tumours	in the CT26 model, 80% of mice were completely cured; the synergistic effect of IL-24 and GM-CSF increased IFN-γ production	[171]
IFNβ	TK-/B18R-/IFN-beta+; JX-795 (WR vvDD)	murine colorectal adenocarcinoma CMT-93 and murine mammary adenocarcinoma JC; C57/BL6 and Balb/c mice respectively	a single intratumoural injection of a high dose of the virus resulted in complete tumour regression. Intravenous injection of the same dose was much less efficient. A significant increase in the tumour-infiltrating lymphocytes was found in all treated animals. Animals with complete responses, showed protection to tumour cell rechallenge	[37]
SIRPα-Fc	SIRPα-Fc-VV (VSC20–WR vvDD)	human osteosarcoma LM7 SCID-Bg mice xenograft model, murine F420 osteosarcoma model; C57BL/6	induced phagocytosis of tumour cells by M1 as well as M2 macrophages in vitro; macrophages and monocytes recruitment into tumours in vivo; increased survival	[172]
**Rhabdoviruses**				
CD40-L	VSV-CD40L	B16 melanoma in C57BL/6 mice	infection with VSV-CD40L induced maturation of bone marrow-derived DCs with increase in expression of CD40, CD86, and MHC II compared to VSV-GFP; some mice showed complete response, however, there was no difference in the anti-tumour response between the control VSV-GFP and VSV-CD40L; no tumour specific antigen response observed	[173]
Flt3L (soluble Fms-like tyrosine kinase 3 ligand)	VSV-Flt3L	murine tumour VSV-resistant B16 melanoma and VSV-sensitive E.G7 T lymphoma	modest animal survival in E.G7 tumour model was independent of adaptive CTL response; tumour-associated DCs were actively infected by VSV in vivo, which prevented their migration and antigen presentation	[50]
IFNγ	VSVΔ51-IFNγ	4T1 mammary carcinoma and CT26 colon carcinoma murine models	VSVΔ51-IFNγ induced secretion of pro-inflammatory factors in the blood, enhanced activation of DCs, and generated a greater tumour-specific immune response; the reduction in tumour size correlated with prolonged survival	[174]
IFNβ-NIS (sodium iodide symporter)	VSV-mIFN bβ-NIS	syngeneic murine acute myeloid leukaemia (AML) C1498 tumour C57 BL/6J PD-L1Ab (10F.9G2; BioXCell)	combination with anti-PD-L1 therapy enhanced anti-tumour activity and survival compared with treatment with virus or antibody alone; increased tumour-infiltrating CD4^+^ and CD8^+^ cells; depletion of CD8 or natural killer cells, but not CD4 cells, resulted in a loss anti-tumour activity	[49]
**Alphaviruses**				
GM-CSF	SFV-GM-CSF	murine i.p. growing ovarian tumour (MOT) spontaneous teratocarcinoma of C3HeB/FeJ mice	single i.p. injection of SFV-GM-CSF leads to increase in the number of peritoneal macrophages and neutrophils. Tumour growth delay for 2 weeks did not lead to prolonged survival.	[175]
IFNγ	SFV-enh/IFNγ	4T1 murine mammary carcinoma spheroids; Balb/c mice orthotopic and s.c. tumours	significant inhibition of tumour growth in comparison to the control SFV/Luc virus; increased CD4^+^ and CD8^+^ cell populations, and decreased T-reg and myeloid CD11b^+^, CD38^+^, and CD206^+^ cell populations in treated tumours	[176]
Flt3L (a soluble Fms-like tyrosine kinase 3 ligand) and XCL1 (a chemoattractant for cDC1 cells)	SFV-XCL1-sFlt3L (SFV-XF)	murine colon cancer MC38 and B16-OVA tu-mours	delayed progression of tumours; increased infiltration of CD8+ T cells and enhanced anti-tumour activity of BATF3-dependent cDC1; the SFV therapeutic activity was potentiated by combination with anti–PD-1, anti-CD137, or CTLA-4 antibodies	[177]
**Reoviruses**				
GM-CSF	rS1-mmGMCSF (MRV)	murine model of pancreatic cancer	intratumoral treatment led to activation of DCs and T-cells	[178]
**Paramyxoviruses**				
GM-CSF	NDVhuGM-CSF, MEDI5395 NDVmuGMCSF (NDV)	176 human tumour cell lines, patient-derived triple-negative breast cancer (TNBC) xenograft model, CT26 murine colon carcinoma tumour model	MEDI5395 has oncolytic and immune stimulating activity in a range of human tumour models, with the most sensitive HT1080, DU145, CAL27, NCIH358, and OVCAR4 cells. Tumour treatment led to inflamed TME, efficacy was further improved by combination with CPIs or T-cell agonists	[179]
MIP3α—(CCL20)	NDV-MIP3α (NDV)	B16 melanoma and CT26 colon carcinoma tumour-bearing mice	enhanced anti-tumour activity; attraction of DCs and induction of adaptive immunity	[180]

**Table 3 biomedicines-10-02142-t003:** Virus vectors encoding cytokines and other molecules activating NK and T cells.

Transgene	Virus Vector (Virus Backbone)	Model	Effect	Ref.
**Adenoviruses**				
IL-2 (modified)	Ad5/3-E2F-d24-vIL2 (Ad5/3)	pancreatic ductal adenocarcinoma (PDAC), hamsters	efficient anti-tumour response (62.5%); reversed immunosuppression by a decrease of myeloid cell populations and an increase of tumour-infiltrating CTLs.	[189]
IL-12 plus TK suicide gene	yCD/mutTKSR39rep-hIL-12 (Ad5)	12 patients with metastatic pancreatic cancer (T2N0M1-T4N1M1)	good toxicity profile; induced immune activation and improved survival; elevated IL-12, IFNγ and CXCL10 serum levels were detected in 42%, 75%, and 92% of patients, respectively	[190]
IL-18	ZD55-IL-18 (Ad5)	human A375 melanoma; nude mouse xenograft model	ZD55-IL-18 and dacarbazine drug (DTIC) showed synergistic effects and resulted in significant tumour cell apoptosis, decreased VEGF expression and inhibition of lung metastasis	[191]
IL-12 plus IL-18	RdB/IL-12/IL-18 (Ad5)	B16-F10 murine melanoma	improved anti-tumour effects, as well as increased survival; elevated levels of IL-12, IL-18, IFNγ and GM-CSF, and infiltration of NK cells, CD4^+^ and CD8^+^ T cells in treated tumours	[192]
anti-PD-L1 blocking antibody	HCA-EFZP-aPDL1 (Ad5)	murine colon carcinoma MC38 tumour	significant reduction of tumour growth with minimal release of the antibody to the bloodstream	[193]
**Herpesviruses**				
IL-12	R-115 (hHER2 retargeted ΔICP34.5 oHSV)	human breast cancer SK-OV-3 cells, Lewis lung carcinoma murine cells expressing hHER2 (HER2-LLC1) s.c. tumours	R-115 reversed the immunosuppressive TME by induction of immunomodulatory cytokines, including IFNγ, promotion of Th1 polarization, and generation of durable responses in some treated animals	[194]
IL-12	M002 (ΔICP34.5 oHSV1)	murine ovarian adenocarcinoma: ID8, Ig10, M0505, and STOSE	reduced peritoneal metastases and improved survival after a single intraperitoneal injection	[195]
IL-12	M032 (ΔICP34.5 oHSV1)	*A. nancymae* monkeys	toxicology and biodistribution study; the protocol for phase I clinical trial in patients with recurrent or progressive malignant glioma was designed	[196]
B7.1-Ig IL-12 IL-18	vHsv-B7.1-Ig, vHsv-IL-12, vHsv-IL-18 (oHSV-1 G47Δ)	murine neuroblastoma Neuro2a, s.c. tumours in A/J mice	the most significant anti-tumour effect by treatment with all three viruses; the effect is abrogated in immune-deficient nude mice, proving the specific T cell-mediated tumour regression	[197]
PDL1 BiTE (anti-PD-L1 scFv plus anti-CD3 scFv)	oHSV-1 PD-L1 BiTE (oHSV-1 G207 backbone)	patient-derived ascites model	the endogenous T cells within malignant ascites were activated generating a pro-inflammatory response and eliminating PD-L1-positive tumour cells and macrophages	[198]
**Poxviruses**				
IL-12	vvDD-IL-12-FG (WR strain of VV-VSC20)	murine colon adenocarcinoma MC38-luc, CT26-luc, and lung mesothelioma AB12-luc	potent anti-tumour effects with complete regression of tumours and re-challenge protection; vvDD-IL-12-FG synergised with anti-PD-1 antibody treatment leading to the cure of all late-stage MC38 tumours; tumour analysis showed a decrease in Tregs, TGF-β, COX-2, VEGF and increase in infiltration by CD8^+^ and CD4^+^ T expressing IFNγ	[199]
CD-80 (B7.1)	rV-B7.1 (VV Wyeth strain)	phase I clinical trial, 12 melanoma patients	partial response was observed in one patient; disease stabilization in two patients; tumour regression was associated with increased expression of CD8 and IFNγ	[200]
CD-80 (B7.1), ICAM-1, LFA-3	rF TRICOM T-cell costimulatory molecules (fowlpoxvirus)	phase I clinical trial, 10 melanoma patients, 2 colon adenocarcinoma	Well-tolerated treatment; however, limited tumour-specific T cell responses; all patients exhibited anti-viral antibody responses	[201]
IL-21	VVLDTKDN1L-mIL-21 (WR strain of VV)	murine colorectal cancer CMT93, s.c. tumours	enhanced anti-tumour immune response able to eliminate primary tumours; induction of systemic anti-tumour immunity preventing tumour recurrence	[202]
anti PD-L1 scFv (single-chain variable fragment) hNIS (human sodium iodide symporter)	CF33-hNIS-anti-PD-L1 (CF33- chimeric poxvirus)	xenograft model of triple-negative breast cancer TNBC (MDA-MB-468)	completely control of tumour growth at low dose	[203]
CCL5 (RANTES)	vvCCL5 (WR vvDD)	murine colon carcinoma M38 s.c. tumours, C57/BL mice	significant tumour suppression and enhanced survival	[204]
**Rhabdoviruses**				
IL-12	rVSV-IL12 (VSV)	murine squamous cell carcinoma (SCC), orthotopic model in C3H/HeJ mice	significant reduction in tumour volume and substantial survival benefits	[205]
IL-15	VSV-IL-15 (VSV)	murine colon carcinoma CT-26 tumour	enhanced anti-tumour T-cell responses and improved survival	[206]
IL-15	VSV-IL-15 (VSVΔM51)	Panc02 murine pancreatic ductal carcinoma, C57BL/6 mice s.c. and orthotopic tumours	VSV-IL-15 was superior over VSV-GFP control in combination with NKT cell therapy; significant tumour regression and increase in survival; the addition of anti PD1 therapy induced complete regressions in 20% of treated animals	[207]
**Paramyxoviruses**				
IL-2	NDV/Anh-IL-2 (NDV; Anhinga strain)	murine H22 hepatocellular carcinoma	Efficient inhibition of tumour growth; 60 days post-treatment, mice which were completely cured were protected against rechallenge with the same tumour cells	[208]
IL-12 PD-L1 antibody	MeVac FmIL-12 MeVac anti-PD-L1 (MV)	MC38cea murine colon carcinoma model	Th1 cell-directed response was revealed by secretion of IFNγ, TNFα and activation of NK and CTLs, leading to complete tumour regression in 90% of treated animals.	[209]
**Alphaviruses**				
IL-12	SFV-IL-12 SFVenh-IL-12 (SFV)	MC38 murine colon adenocarcinoma	≥80% complete tumour regression with potent CTL responses and long-term tumour-free survival; improved efficiency was shown by repeated intratumoral injections.	[210]
PD-L1 antibody	SFV-αPDL1 (SFV)	MC38 murine colon adenocarcinoma	>40% complete regression compared with less efficient AAV-αPDL1and αPDL1 monoclonal antibodies given systemically or locally	[211]

**Table 4 biomedicines-10-02142-t004:** Virus vectors encoding cytokines and other molecules for remodelling tumour stroma and multiple immune cell populations.

Transgene	Virus Vector (Virus Backbone)	Model	Effect	Ref.
**Adenoviruses**				
sTGFβRIIFc	rAd.sT	murine breast cancer 4T1 and renal cancer Renca tumours; Balb/c	inhibition of both tumour growth and lung metastases; induction of Th1 immune response with CTL tumour infiltration, promotion of CD4^+^ T memory cells, reduction of Tregs, and bone marrow-derived suppressor cells	[213]
Hyaluronidase PH20	AdwtRGD-PH20 ICOVIR17	human melanoma SkMel-28 xenografts; nude mice	degradation of hyaluronan, enhanced viral distribution, and tumour regression. The anti-tumour activity of replication-competent and tumour-selective ICOVIR17 was higher in comparison to AdwtRGD-PH20 virus	[214]
Hyaluronidase	VCN-01 (Ad5)	osteosarcoma patient cell lines (531MII, 678R, 588M, 595M) and a commercial cell line (143B); nude mice xenografts	potent anti-sarcoma effect in vitro and in vivo in mouse models of intratibial and lung metastatic osteosarcoma, with complete tibial tumour regression in the high dose (10^8^ pfu) group	[215]
TNFα IL-2	Ad5-CMV-mTNFα Ad5-CMVmIL2 (Ad5)	murine B16.OVA melanoma; C57 BL/6JOlaHsd mice	complete tumour regression in all animals treated with anti-PD1 antibodies and corresponding viruses; Th1 immune response and increased intra-tumoral proportion of CD8^+^ and CD4^+^ T cells	[6]
GM-CSF CTLA-4 ab	SKL001 SKL002 Ad5	CMT-64 mouse small lung carcinoma, B16F10 murine melanoma, human A549 lung s.c. xenograft model	selective replication and anti-tumour activity after intravenous administration was shown in mouse B16F10 melanoma tumour and human tumour xenograft model; combination of the viruses potentiated anti-tumour activity	[216]
Anti TGFβshRNA GM-CSF (in one vector) plus MART1 (DNA/TAA)	AdGshT	murine B16BL6-CAR/E1B55 malignant melanoma; C57BL/6 mice	treatment by both DNA vaccine expressing TAA (MART1) and oncolytic adenovirus, encoding GM-CSF together with shRNA to TGF-β2 resulted in significant anti-tumour effects, however, complete regression of tumours was not achieved	[217]
IL-12p35, IL-12p40; GM-CSF and RLX (relaxin)	oAd/RLX oAd/IL12/GM-RLX	Syrian hamster s.c. and orthotopic pancreatic tumour models.	expression of IL-12, GM-CSF and RLX mediated by a single oncolytic Ad vector promoted remodelling of TME to potentiate antibodies-based therapies	[218]
IL-12 plus VEGF binding shRNA	RdB/IL12/shVEGF (Ad5)	murine B16-F10 melanoma; C57BL/6 mice	Efficient anti-tumour effect with massive tumour infiltration of differentiated CD4^+^ T cells, CD8^+^ T cells, NK cells, and DCs. Suppressed expression of VEGF, supporting the restoration of the anti-tumour immune response	[219]
**Herpesviruses**				
CCL2 mIL-12	M010 M002 (ΔICP34.5 oHSV)	neuroblastoma Neuro-2a tumours s.c. syngeneic A/J mouse strain	combined treatment led to the most efficient tumour growth inhibition	[220]
IL-12 IL-15 PD1v GM-CSF IL7 plus CCL19	oHSV2-IL12, -PD1v, -IL15, -IL7-CCL19, -GM-CSF ΔICP34.5 ΔICP47 oHSV2 (HG52 strain)	breast cancer 4T1 and colon carcinoma CT26 murine tumour models; Balb/c mice	all vector variants used as a single treatment have had a similar anti-tumour activity; the most potent activity was demonstrated for all five virus vector combinations; the tumour re-challenge exhibited that cocktail therapy prevents secondary tumourogenesis	[221]
IL-12 GM-CSF (in one vector)	R-123 hHER2 retargeted ΔICP34.5 oHSV	human breast cancer SK-OV-3 cells, Lewis lung carcinoma murine cell line expressing hHER2 (HER2-LLC1) s.c. tumours; hHER2-transgenic C57BL/6 mice (B6.Cg-Pds5bTg(Wap ERBB2)229Wzw/J)	combined treatment with anti-PD1 led to significant inhibition of tumour growth with complete tumour resection in case, (mGM-CSF), mIL-12+mGM-CSF) expressing vector; systemic delivery of double-armed virus combined with anti-PD1 inhibited the development of tumour metastasis	[222]
**Poxviruses**				
PD-1 fused with IgG1 Fc plus GM-CSF	VV-iPDL1/GM WR vvDD	murine s.c. tumours Luc B16-F10 melanoma; Murine breast cancer Py230 and MC38 colon adenocarcinoma; C57BL/6 mice	the highest tumour growth inhibition was observed in VV-iPDL1/GM treated animals, compared to single treatments; CD8 T cell depletion significantly abolished the systemic anti-tumour activity of VV-iPDL1/GM; increased DCs (CD11c^+^) infiltration was observed in VV-iPDL1/GM treated mice	[3]

## Data Availability

Not applicable.

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
