# Peer review of "Recombinant Viral Vectors for Therapeutic Programming of Tumour Microenvironment: Advantages and Limitations"

_biomedicines, 2022, doi:10.3390/biomedicines10092142_

Round 1

Reviewer 1 Report

The article by Spunde et al. is a very extensive review about the use of recombinant viral vectors for antitumor therapy. The authors have made a very laborious and detailed revision of all the different strategies based on viral vectors, including replicative and non-replicative viruses, to activate immune responses against tumors. The manuscript is well written, and the authors have also included very informative tables where it is easy to check which vectors have been used, which transgenes were expressed, and what therapeutic effects were observed. My only criticism is the length of the review, maybe justified by the wide scope of the topic.

Some minor points should be addressed before publication:

- Line 37: Where it is written: “Despite to genetic instability”, it should be: “Despite genetic instability”.

- Line 57. I would not consider T-VEC as a cancer vaccine since it is not expressing any tumor antigen, please correct.

- Line 255. When the authors wrote: “ that SFV1 vectors”, what do they mean by SFV1? 

- Line 598. The authors wrote : “4-1BB is expressed on B cells, macrophages, and DCs, whereas its ligand 4-1BBL is expressed on DCs and macrophages”. This is correct but 4-1BB is also expressed on activated T and natural killer (NK) cells and this should be mentioned.

- Line 776. When talking about vectors expressing rFlt3L the authors forgot to mention a relevant publication expressing this growth factor with a SFV vector: Sánchez-Paulete AR, et al I. Cancer Res. 2018 Dec 1;78(23):6643-6654. doi: 10.1158/0008-5472.CAN-18-0933.

- Line 1152: The authors wrote: “Ad5 vectors SKL001 SKL002 encoding GM-CSF and CTLA-4, respectively” But the vector expresses in fact an antibody against CTLA-4, please specify it.

Line 1245. “is demonstrated absolutely…” should be “has demonstrated absolutely…”

Author Response

(Reviewer 1)

Dear Editor and reviewers,

Thank you very much for your careful work in reading and editing our manuscript ”Recombinant Viral Vectors for Therapeutic Programming of Tumour Microenvironment: Advantages and Limitations”.

We have revised the manuscript according to the reviewers’ comments and suggestions. Below, please, find our answers and step-by-step explanations of the changes which have been introduced into the manuscript. The original text of the reviews is given in blue.

We hope that you will find the revised version suitable for publication in “Biomedicines”.

Best regards

Sincerely yours

Anna Zajakina,

acting on behalf of all co-authors

The article by Spunde et al. is a very extensive review about the use of recombinant viral vectors for antitumor therapy. The authors have made a very laborious and detailed revision of all the different strategies based on viral vectors, including replicative and non-replicative viruses, to activate immune responses against tumors. The manuscript is well written, and the authors have also included very informative tables where it is easy to check which vectors have been used, which transgenes were expressed, and what therapeutic effects were observed. My only criticism is the length of the review, maybe justified by the wide scope of the topic.

We agree that the review is rather extensive and large, because broad aspects, covering viral vectors, TME composition, and vector-based treatment strategies have been highlighted. Nevertheless, we believe that the structure of the review and the tables and figures provided make it easy to understand and follow the data presented. The classification of vectors is given and the respective transgenes are categorized with respect to their immunomodulating targets. Despite the large volume of the review, making it laborious for reviewers and authors, we hope the manuscript will be interesting to a wide audience, providing a general understanding of the achievements and bottlenecks in the promising field of virus-based TME programming.

Some minor points should be addressed before publication:

- Line 37: Where it is written: “Despite to genetic instability”, it should be: “Despite genetic instability”.

was corrected (line 37)

- Line 57. I would not consider T-VEC as a cancer vaccine since it is not expressing any tumor antigen, please correct.

We fully agree, was corrected (line 57)

- Line 255. When the authors wrote: “ that SFV1 vectors”, what do they mean by SFV1? 

“ that SFV1 vectors” was changed to “replication-deficient SFV vectors”. SFV1 is the type of replication-deficient SFV vector. We do not explain the details of the vector structure (it is provided in the respective reference), therefore, the general expression “replication-deficient SFV vectors” is more relevant.

was corrected (line 272)

- Line 598. The authors wrote : “4-1BB is expressed on B cells, macrophages, and DCs, whereas its ligand 4-1BBL is expressed on DCs and macrophages”. This is correct but 4-1BB is also expressed on activated T and natural killer (NK) cells and this should be mentioned.

was specified (line 612)

- Line 776. When talking about vectors expressing rFlt3L the authors forgot to mention a relevant publication expressing this growth factor with a SFV vector: Sánchez-Paulete AR, et al I. Cancer Res. 2018 Dec 1;78(23):6643-6654. doi: 10.1158/0008-5472.CAN-18-0933.

The respective description and reference, including in Table 2, were provided (lines 816-823).

“More promising results were obtained with alphavirus SFV vector simultaneously expressing a soluble Flt3L and an XCL1 chemoattractant for classical DC1 (cDC1) cells [177]. Repeated intratumoral injection of the vector led to the delayed progression of syngeneic murine colon cancer MC38 and B16-OVA tumours. The treatment increased the infiltration of CD8+ T cells and facilitated the anti-tumour activity of BATF3-dependent cDC1 in tumour-bearing mice. Furthermore, the SFV therapeutic activity was potentiated by combination with anti–PD-1, anti-CD137, or anti-CTLA-4 immunomodulating antibodies.”

Some more details were also provided for VSV Flt3L study (lines 807-812).

- Line 1152: The authors wrote: “Ad5 vectors SKL001 SKL002 encoding GM-CSF and CTLA-4, respectively” But the vector expresses in fact an antibody against CTLA-4, please specify it.

was corrected (line 1204)

Line 1245. “is demonstrated absolutely…” should be “has demonstrated absolutely…”

was corrected (line 1316)

Some more minor corrections were made, and additional references were added.

Reviewer 2 Report

The review describes current viral vector platforms and transgenes used in these platforms to modulate the tumour microenvironment. The review is quite extensive and timely as there are multiple viral platforms reaching to clinical studies.

Comments that need to be properly addressed:

-Although this is a review and not an opinion letter, there are surprisingly many part in the text that seem to be without citation, why is this?

-line 180 sentence starting Replication deficient.. is missing a reference (the whole chapter is missing a citation).

-Chapter 3.1.3 Herpesviruses authors need to include teserpaturev (G47∆; Delytact) to the text as this is the first oncolytic virus to receive approval for use in brain cancer.

-Table 1 Adenovirus insert capacity is 7.5-36 kb, this is good to define that the 36kb is for gutless vectors

-Table 1 Adenovirus limitations there are mentioning (without refs) of biosafety concerns (random integration), this needs to be discussed and referenced. How is this seen as a concern, is there many examples of this? Ad vectors has been given to millions of people (AZD1222) without this concern (although other concerns have emerged).

-Table 1 Reovirus limitations there is a bullet point: very insert capacity

-Chapter 7.1 GM-CSF expression, the authors mention T-VEC and Pexa-Vec trials but do not mention the recent clinical trial failures of these two platforms. These needs to be critically mentioned in this review.

-The authors do not mention one of the most potential ways of modulating the TME that is increasing the tumour-specific T cell number by virus-delivered tumour antigens. The pioneering work by Vincenzo Cerullo (coating viruses with tumour antigens) and John Bell (expressing tumour antigens from the viral platform) needs to be addressed here. Both of these strategies have entered into clinical trials (NCT05492682, NCT02285816 and NCT03618953) also this trial and the platform is worth mentioning NCT05141721.

-Also, it is worth mentioning Nadofaragene firadenovec (rAd-IFNa/Syn3) that expresses INF alfa-2b as this Ad-based therapy has successfully completed phase III studies and is currently applying marketing authorization from FDA.

-The authors note that pre-existing immunity towards vector platform may impact negatively to the efficacy of the platform although there are multiple studies describing the opposite effect, this could be briefly discussed in the text.

Author Response

(Reviewer2)

Dear Editor and reviewers,

Thank you very much for your careful work in reading and editing our manuscript ”Recombinant Viral Vectors for Therapeutic Programming of Tumour Microenvironment: Advantages and Limitations”.

We have revised the manuscript according to the reviewers’ comments and suggestions. Below, please, find our answers and step-by-step explanations of the changes which have been introduced into the manuscript. The original text of the reviews is given in blue.

We hope that you will find the revised version suitable for publication in “Biomedicines”.

Best regards

Sincerely yours

Anna Zajakina,

acting on behalf of all co-authors

The review describes current viral vector platforms and transgenes used in these platforms to modulate the tumour microenvironment. The review is quite extensive and timely as there are multiple viral platforms reaching to clinical studies.

Our review is rather extensive and large, because broad aspects, including viral vectors, TME composition, and vector-based treatment strategies have been highlighted. We were mostly focused on the expression of cytokines and soluble factors for TME programming. Therefore, some aspects (TAA vaccines, oncolytic vectors without transgenes, as well as integrating vectors AAV, RV) were not covered and remained out of the scope of this manuscript. We believe that the structure of the review and the tables and figures provided make it easy to understand and follow the data presented. The classification of vectors is given and the respective transgenes are categorized with respect to their immunomodulating targets. Despite the large volume of the review, making it laborious for reviewers and authors, we hope the manuscript will be interesting to a wide audience, providing a general understanding of the achievements and bottlenecks in the promising field of virus-based TME programming.

 Comments that need to be properly addressed:

-Although this is a review and not an opinion letter, there are surprisingly many part in the text that seem to be without citation, why is this?

We revised the text and added more key references. Totally, 27 new references were added (marked in yellow).

One of the reasons, why some citations were missing in sections related to viral platforms and tumour microenvironment is that we have referred also to specialized reviews, which may provide detailed information on respective issues.

-line 180 sentence starting Replication deficient.. is missing a reference (the whole chapter is missing a citation).

Lines 189, 192

two references were added

-Chapter 3.1.3 Herpesviruses authors need to include teserpaturev (G47∆; Delytact) to the text as this is the first oncolytic virus to receive approval for use in brain cancer.

Lines 219-228.

The details on the development of G47∆ vector and its clinical use were provided, including respective references.

“Deletion of neurovirulence factor ICP34.5 attenuates HSV viral pathogenicity allowing the virus to replicate selectively in tumours. Furthermore, the HSV1 vector carrying combined deletions in ICP34.5 and ICP6 (large subunit of ribonucleotide reductase) - G207, was engineered, allowing virus replication in human glioma cells. However, deletion of ICP34.5 attenuated virus replication in glioma cells. G207 backbone vector was further modified by deletion of the ICP47 gene. The resulting triple-mutated G47∆ vector showed improved efficiency in glioblastoma tumours in comparison to the parent G207 vector [41]. In addition, ICP47 deletion reduces virus-mediated suppression of antigen presentation. The G47-based therapy for glioblastoma - teserpaturev Delytac - recently received approval for treatment of GBM in Japan [44]”

-Table 1 Adenovirus insert capacity is 7.5-36 kb, this is good to define that the 36kb is for gutless vectors

was specified

The respective explanation was added in the text (lines 133-138) and in Table 1.

“Last generation helper-dependent adenovirus vectors (HDAds), also called “gutless” or fully deleted vectors, containing only inverted terminal repeats and the genome packaging signal, allow up to 36 kb insertions. Besides high capacity, these third-generation vectors possess also reduced immunogenicity [23].”

-Table 1 Adenovirus limitations there are mentioning (without refs) of biosafety concerns (random integration), this needs to be discussed and referenced. How is this seen as a concern, is there many examples of this? Ad vectors has been given to millions of people (AZD1222) without this concern (although other concerns have emerged).

We have addressed this issue in Summary section discussing the potential limitations of the vectors (lines 1269-1280).

“Although non-integrating DNA viruses replicating in the cell nucleus are proposed to be safe in respect of insertional mutagenesis, the risk of non-specific chromosomal integration of viral DNA should not be ignored, considering the genetic instability and heterogeneity of cancer cells, especially in case of repeated local injection of high titre virus preparations. The experimental setting to evaluate the probability of random integration and points of insertions is rather sophisticated and difficult, because of that such works are rare [226]. Nevertheless, such risks were assessed regarding adenovirus vectors showing, that the probability of heterologous integration of adenovirus vector is higher in comparison to homologous one in murine models [227,228]. Furthermore, in earlier studies, it was observed that injection of hamsters with wild-type adenovirus type 12 (Ad12) led to tumour development due to chromosomal integration of virus DNA and expression of the E1A/E1B oncoproteins [229].”

-Table 1 Reovirus limitations there is a bullet point: very insert capacity

 was corrected “very low insert capacity”

-Chapter 7.1 GM-CSF expression, the authors mention T-VEC and Pexa-Vec trials but do not mention the recent clinical trial failures of these two platforms. These needs to be critically mentioned in this review.

the respective studies were highlighted and shortly discussed

Lines 728-732

“In phase III trial OPTiM, in patients with unresectable melanoma, i.t. administrated T-VEC proved clear survival benefits with complete responses in 16.9% patients versus 0.7% of the control group received s.c. injections of recombinant GM-CSF, with best results in earlier stage metastatic melanoma of lower initial tumour size [162].”

Lines 735-739

“In phase II clinical trial with T-VEC and ipilimumab (NCT01740297) the significant improvement was achieved in combination therapy [163]. However, the results of phase III clinical trial (NCT02263508) with T-VEC and pembrolizumab was disappointing as no significant improvement in survival of patients with advanced melanoma was observed.”

Lines 749-754

“However, in the phase III PHOCUS trial (NCT02562755) enrolling 459 patients with advanced hepatocellular carcinoma, treatment with Pexa-Vec followed by sorafenib was inefficient and showed no improvement compared with the control group. Probably, the use of therapeutic virus is insufficient for treatment of late stages of cancer, which should be addressed in future trials.”

-The authors do not mention one of the most potential ways of modulating the TME that is increasing the tumour-specific T cell number by virus-delivered tumour antigens. The pioneering work by Vincenzo Cerullo (coating viruses with tumour antigens) and John Bell (expressing tumour antigens from the viral platform) needs to be addressed here. Both of these strategies have entered into clinical trials (NCT05492682, NCT02285816 and NCT03618953) also this trial and the platform is worth mentioning NCT05141721.

Our review is focused on TME immunomodulating cytokines and other soluble molecules able to change the immune cell composition/activity and the TME state. Cancer vaccines based on TAA are not covered by this manuscript, there are many specialised reviews highlighting broad aspects of TAA vaccines. However, we agree that key information should be added to the Introduction section. Please see the respective description and references, outlining some important aspects of TAA based vaccines.

Lines 61-67

“Nevertheless, as promising approach cancer vaccines such as TAAs expressing viruses or TAAs coated viral particles continue to enter clinical trials such as MAGE-A3 expressing viruses (NCT02285816), papillomavirus E6E7 antigen expressing viruses (NCT03618953) as well as innovative peptide-coated conditionally replicating adenovirus - PeptiCRAd-1 (NCT05492682) [9]. Significant work has been done to improve vector-based cancer vaccines [10], however, TAA-based vaccines are out of the scope of this review.”

-Also, it is worth mentioning Nadofaragene firadenovec (rAd-IFNa/Syn3) that expresses INF alfa-2b as this Ad-based therapy has successfully completed phase III studies and is currently applying marketing authorization from FDA.

The description on rAd-IFNa/Syn3 vector phase III study was provided in text and in Table 2.

Lines 881-889

“Finally, replication-deficient adenovirus vector - rAd-IFN/Syn3, expressing IFN-2b gene which was combined with a polyamid surfactant (Syn3) enhancing the viral trans-duction of the urothelium [187], entered a phase III study (NCT02773849) for treatment of non-muscle-invasive bladder cancer [188]. After intravesical treatment more than half of patients with carcinoma in situ achieved complete response during 3 months and half of them were free from high-grade recurrence at 12 months. Clinically meaningful responses were achieved also in patients with high-grade Ta or T1 bladder cancer. Currently, the phase III clinical trial is ongoing, overall providing promising results and probably a novel cancer therapy drug based on adenovirus vector will be approved in near future.”

-The authors note that pre-existing immunity towards vector platform may impact negatively to the efficacy of the platform although there are multiple studies describing the opposite effect, this could be briefly discussed in the text.

We agree, a brief discussion was added in Summary section.

Lines 1290-1294

“On the other hand, the innate response against the intratumorally administered virus vector in presence of relevant memory T- cells due to previous exposure to the parental virus can be sufficient to overcome the TME immunosuppressive conditions and boost the efficient adaptive anti-tumour immune response [230,231]. Therefore, the pre-existing immunity towards the vector may promote the immune cell activation.”

Some more minor corrections were made.